



# Optimal tuning of engineering wake models through LiDAR measurements

Lu Zhan, Stefano Letizia, and Giacomo Valerio Iungo

Wind Fluids and Experiments (WindFluX) Laboratory, Mechanical Engineering Department, The University of Texas at Dallas, Richardson, Texas, USA

**Correspondence:** Giacomo Valerio Iungo, 800 West Campbell Rd, Richardson, TX 75080-3021, USA. Email: valerio.iungo@utdallas.edu

**Abstract.** Engineering wake models provide the invaluable advantage to predict wind turbine wakes, power capture and, in turn, annual energy production for an entire wind farm with very low computational costs compared to higher-fidelity numerical tools. However, wake and power predictions obtained with engineering wake models can be not sufficiently accurate for wind-farm optimization problems due to the ad-hoc tuning of the model parameters, which are typically strongly dependent on the

characteristics of the site and power plant under investigation. In this paper, LiDAR measurements collected for individual turbine wakes evolving over a flat terrain are leveraged to perform optimal tuning of the parameters of four widely-used engineering wake models. The average wake velocity fields, used as reference for the optimization problem, are obtained through a cluster analysis of LiDAR measurements performed under a broad range of turbine operative conditions and incoming wind characteristics. The sensitivity analysis of the optimally-tuned model parameters and the respective physical interpretation are presented. The performance of the optimally-tuned engineering wake models is discussed, while the results suggest that

the optimally-tuned Bastankhah and Ainslie wake models provide very good predictions of wind turbine wakes. Specifically, the Bastankhah wake model should be tuned only for the far-wake region, namely where the wake velocity field can be well-approximated with a Gaussian profile in the radial direction. In contrast, the Ainslie model provides the advantage of using as input an arbitrary near-wake velocity profile, which can be obtained through other wake models, higher-fidelity tools or experimental data. The good prediction capabilities of the Ainslie model indicate that the mixing-length model is a simple,

yet efficient, turbulence closure to capture effects of incoming wind and wake-generated turbulence on the wake downstream evolution and predictions of turbine power yield.

## 1 Introduction

Wake interactions are responsible for significant power losses of wind farms (Barthelmie, et al., 2007; El-Asha et al., 2017) and,

thus, numerical tools for predicting the intra-wind-farm velocity field are highly sought for the optimal design of wind farm layout (Kusiak et al., 2010; González et al., 2010; Santhanagopalan et al., 2018b), develop control algorithms for improving turbine operations (Lee et al., 2013; Annoni et al., 2016), and enhancing accuracy in predictions of power capture (Tian et al., 2017).





Wind turbine wakes present a structural paradigm where the flow responds to both turbine settings and incoming wind conditions: the former being associated with thrust coefficient affecting power production and wake velocity deficit (Iungo et al., 2018a), while the latter being a combination of velocity variability within the atmospheric boundary layer and turbulence characteristics depending on the atmospheric stability regime (Hansen et al., 2012; Iungo et al., 2014).

Engineering wake models have widely been used in the wind energy industry because providing a good trade-off between fidelity, in terms of accuracy of the predicted flow and turbine power capture, and required computational costs. High fidelity models, such as large-eddy simulations (LES), enable detailed characterization of the wake flow and dynamics, together with effects on wind turbine performance (Martínez-Tossas et al., 2015; Santoni et al., 2017). However, the required high computational costs make LES not a suitable tool for wind farm optimization problems. On the other hand, mid-fidelity models have been proposed as tools to bridge the need to resolve flow physics with adequate spatio-temporal resolution and the constraint of achieving results in a timely manner. Among mid-fidelity wake models, we would mention the dynamic meandering wake model (Larsen et al., 2007), prescribed vortex wake model (Chattot, 2007; Shaler et al., 2019), free vortex wake model (Sebastian et al., 2012) and data-driven RANS models (Iungo et al., 2015, 2018a; Santhanagopalan et al., 2018a).

For wind farm problems involving hundreds to thousands of simulations, engineering wake models represent suitable tools to achieve predictions of power capture from a wind turbine array in a limited amount of time (Mortensen et al., 2011; Acker et al., 2011). There are two general classes of wake engineering models: the kinematic models (explicit models) solving the conservation of mass and momentum as governing equations to obtain an explicit analytical formulation, while the field models (implicit models) generate predictions of the wake velocity field through a numerical approach.

The pioneering work by Jensen (1983) and Katic et al. (1987) assumed a linear wake expansion and a top-hat shape of the wake velocity profile at each downstream location. Despite its simplicity, this model provides a good estimation of the mean kinetic energy content available for downstream turbines. Based on the Jensen model, Frandsen kept the top-hat wake profile and derived an asymptotic equilibrium state for an infinite wind farm by solving both mass and momentum budgets (Frandsen et al., 2006). More recently, an axisymmetric Gaussian wake velocity distribution has been considered as a wake model (Bastankhah et al., 2014), based on the classical theory for shear flows (Tennekes et al., 1972). For the Bastankhah wake model, the Gaussian distribution of the velocity at a given downstream location has been embedded into the derivation of the mass and momentum budgets. The model parameters were tuned by leveraging datasets obtained through LES and wind tunnel experiments. Furthermore, this Gaussian wake model still inherits the assumption of linear wake expansion from the Jensen model. A more recent model, denoted as one-parameter model, stemmed from the entrainment hypothesis, which is formulated without linearizing the governing equations or assuming the linearity in the growth of the wake width (Luzzatto, 2018).

Other wake models have been developed without assuming the wake velocity distribution and expansion formula. For instance, Larsen used the Reynolds-averaged Navier-Stokes (RANS) equations with the mixing length model as turbulence closure (Larsen, 1988; Swain, 1929). Despite the relatively complicated calibration process, the Larsen wake model has ambient turbulence level embedded in the model.

Field models (implicit models) use different methodologies to resolve the governing equations implicitly. Ainslie developed one of the classic field models and calculated the complete flow field numerically by solving the RANS equations with a



turbulence closure based on the mixing length assumption (Ainslie, 1988). This model has identical governing equations as for
the Larsen model, for which the turbulent eddy viscosity is modeled based on the wake-generated and ambient turbulence. For
the Ainslie model, the governing equations are solved numerically with a parabolic approach to reduce computational costs.

Engineering wake models generally require parametric calibration. Light detection and ranging (LiDAR) measurements
were used to calibrate and validate the wake growth-rate of the Bastankhah wake model obtaining a good agreement between
the model predictions and the experimental data (Carbajo et al., 2018). Wind tunnel experiments were performed to improve
the initial profile and filter function of the Ainslie model (Kim et al., 2018). For the Larsen model, a calibration procedure is
introduced in the European standards (Dekker, 1998), while it is also calibrated by a data-driven method with high-frequency
SCADA data in Göçmen et al. (2018). Similarly, the SCADA data can be leveraged to tune the wake decay coefficient for the
Jensen model for different values of the incoming wind turbulence intensity (Duc et al., 2019). Several researchers have vali-
dated various wake models for different wind farms showing the non-trivial process to assess accuracy, robustness, reliability,
and uncertainty of the models (Archer et al., 2018; Duckworth et al., 2008; Jeon et al., 2015; Gaumond et al., 2012; Göçmen
et al., 2018). A cluster analysis of the experimental datasets, such as considering atmospheric conditions, turbine settings and
wake velocity fields, has been recommended for wake model benchmarking (Doubrawa et al., 2019).

In this paper, we optimally tune parameters of four engineering wake models based on LiDAR measurements collected for
a utility-scale wind farm. Firstly, the models and their parameters are examined and discussed. The optimization of the model
parameters is carried out by minimizing the objective function defined by the percentage error between the average velocity
field measured by the LiDAR and the respective one predicted by the engineering wake models. The optimal tuning of the
model parameters is performed for various clusters of the LiDAR dataset based on the turbine thrust coefficient and incoming
wind turbulence intensity at hub height. The optimally-tuned parameters of the engineering wake models are then scrutinized
through a linear-regression analysis. Limitations and advantages of the various models will be discussed based on the results
obtained from the optimal calibration process.

The remainder of this paper is organized as follows: the wind farm under investigation and LiDAR dataset are described in
section 2. The four engineering wake models used for the present work are then elucidated together with the methodology of
their optimal tuning in section 3. Finally, the results and discussion of the wake models are reported in section 4, followed by
concluding remarks in section 5.

## 2   LiDAR experiment for a wind farm on flat terrain

A LiDAR experiment was carried out at a wind farm in North Texas made of 39 2.3-MW wind turbines with rotor diameter, $D$,
of 108 m and a hub height of 80 m. The topography map of the site was downloaded from the U.S. Geological Survey (usgs.gov
, url) (Fig. 1(a)), while the meteorological data indicated a prevailing southerly wind direction (Fig. 1(b)). Meteorological
data were provided as 10-minute averages and standard deviation of wind speed, wind direction , temperature, humidity, and
barometric pressure. SCADA data were provided as ten-minute averages and standard deviation of wind speed, power output,



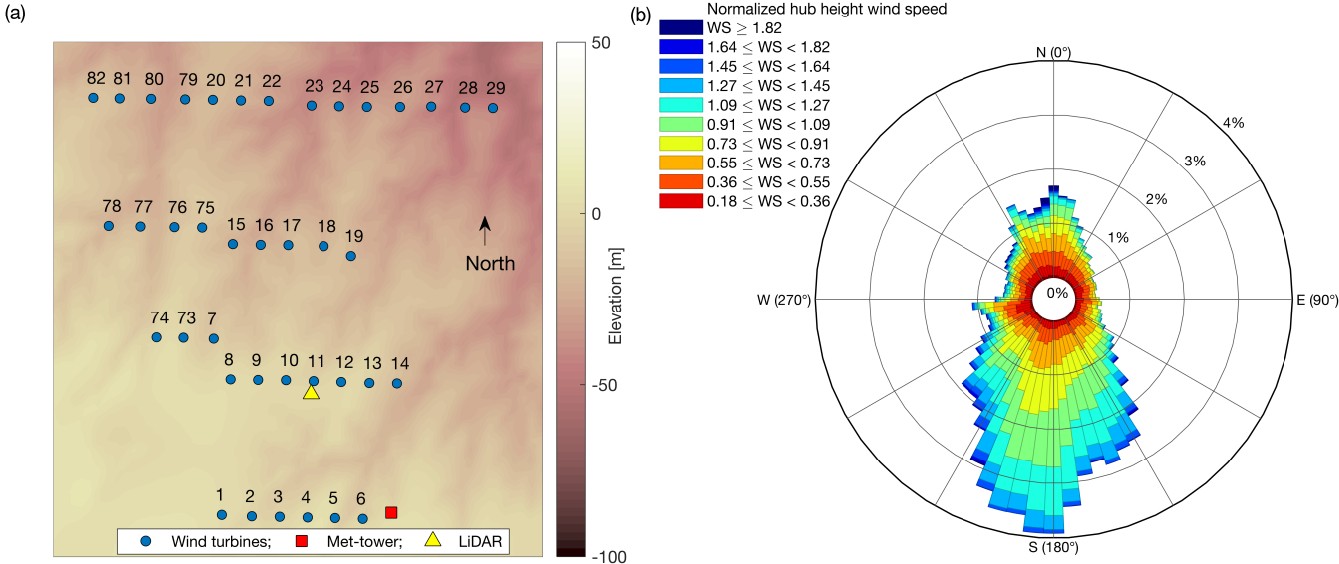

**Figure 1.** Characterization of the test site: a) layout of the wind farm, where the size of the blue markers represents down-scaled rotor diameter; b) wind rose of the hub-height wind measured by the met-tower and reported as a ratio of the turbine rated wind speed.

rotor rotational velocity and yaw angle. For more details on this dataset and the used quality control process see El-Asha et al. (2017) and Zhan, et al. (2019).

The scanning pulsed Doppler wind LiDAR deployed for this experiment is a Windcube 200S manufactured by Leosphere, which emits a laser beam into the atmosphere and measures the radial wind speed, i.e. the velocity component parallel to the laser beam, from the Doppler frequency shift on the back-scattered light. The LiDAR system operates in a spherical coordinate system and measures the radial velocity defined as the summation of three velocity components projected in the laser beam direction. It features a typical scanning range of about 4 km with a range gate of 50 m, accumulation time of 500 ms, and accuracy of 0.5 m/s in wind speed measurements. More details on the LiDAR system and the field campaign are available in Zhan, et al. (2019).

About 10,000 plan-position indicator (PPI) LiDAR scans of isolated wind turbine wakes have been processed to provide the non-dimensional average velocity fields used for this study (Zhan, et al., 2019). LiDAR measurements were clustered according to different categories of inflow condition, namely 13 bins of the hub-height wind speed, and 5 regimes of static atmospheric stability. The various clusters are defined to capture wake variability for different incoming wind turbulence intensity, $TI$, and different turbine operations and, thus, control settings along the turbine power curve. The LiDAR scans within each cluster have been averaged through a technique based on the Barnes scheme (Barnes, 1964; Newsom et al., 2017). For more details on the cluster analysis and the calculation of the ensemble statistics of the LiDAR data see Zhan, et al. (2019), while the clustered statistics are publicly available on Zenodo (Iungo , 2020).



## 3  Data-driven optimal-tuning of engineering wake models

By leveraging the average velocity field of wakes measured with a scanning Doppler wind LiDAR for different atmospheric
stability regimes and rotor thrust coefficients, we perform optimal tuning of four widely-used engineering wake models, namely
the Jensen model (Jensen, 1983), the Bastankhah model (Bastankhah et al., 2014), the Larsen model (Larsen, 1988; Larsen,
G. C. , 2009) and the Ainslie model (Ainslie, 1988). In the following, these models are described, then their parameters are
optimally calibrated based on the LiDAR measurements. Specifically, the objective function of the optimization problem is
the mean percentage error (PE) defined via the LiDAR average streamwise velocity field, $U_{LiDAR}$, and the respective model
prediction, $U_{model}$, as follows:

$$PE = \left\langle \frac{|U_{model}(i,j) - U_{LiDAR}(i,j)|}{U_{LiDAR}(i,j)} \right\rangle, \tag{1}$$

where $\langle \rangle$ refers to the arithmetic mean and $(i,j)$ locates the grid points within the area probed by the LiDAR. The minimization
of $PE$ is performed with a heuristic approach by mapping each model parameter within prescribed ranges.

For this work, it is noteworthy that the thrust coefficient of the turbine rotor can be estimated through the LiDAR measure-
ments, $C_{t_{LiDAR}}$, by leveraging the mass and streamwise-momentum budgets (Iungo et al., 2018b; Zhan, et al., 2019) and from
the SCADA data, referred to as $C_{t_{SCA}}$, which is calculated by applying the actuator disk theory. Considering a 1-D stream-tube
analysis, mass conservation can be expressed as:

$$\rho \pi R_0^2 U_\infty = 2\pi\rho \int_0^R u(r) r dr, \tag{2}$$

where $R_0$ and $R$ are the radius of the bases of the stream-tube upstream and downstream of the turbine rotor, respectively.
The freestream velocity is indicated as $U_\infty$, while $r$ is the radial coordinate, $u$ is the wake streamwise velocity and $\rho$ is the air
density. By neglecting the pressure forces and the shear stresses at the boundary of the stream-tube, the momentum conservation
reads as:

$$\rho \pi R_0^2 U_\infty^2 - 2\pi\rho \int_0^R u(r)^2 r dr = \frac{1}{2}\rho U_\infty^2 \pi \frac{D^2}{4} C_{t_{LiDAR}}. \tag{3}$$

According to Iungo et al. (2018b), this approach can produce a good approximation for the rotor thrust coefficient if the
downstream base of the stream-tube is located in a wake region where the streamwise pressure gradient due to the induction
zone becomes negligible and the turbulent shear stresses are still small compared with those of the far-wake region. By using
this strategy, Eqs. 2 and 3 have been applied by using the LiDAR data acquired at the position $x = 1.75D$ to obtain the two
unknown parameters, $R$ and $C_{t_{LiDAR}}$ for each LiDAR cluster. Specifically, the radius of upstream stream-tube, $R_0$, has been
preset as $0.75D$ in Eq. 2 to determine the downstream radius, $R$, which is then used in Eq. 3 to calculate $C_{t_{LiDAR}}$.
Additionally, the rotor thrust coefficient can be estimated directly from the SCADA data, which is referred to as $C_{t_{SCA}}$. The
streamwise induction factor, $a$, is calculated from the solution of the power coefficient with 1-D stream-tube assumption:

$$\frac{P}{\frac{1}{2}\rho U_\infty^3 \frac{\pi}{4} D^2} = 4a(1-a)^2, \tag{4}$$





where $P$ is the power yield recorded by the SCADA. Subsequently, the thrust coefficient is calculated as follows:

$$C_{t_{SCA}} = 4a(1-a). \tag{5}$$

These two parameters, $C_{t_{LiDAR}}$ and $C_{t_{SCA}}$, allow us to gauge the accuracy in the optimal tuning of the engineering wake models.

### 3.1 Jensen wake model

For the Jensen wake model, mass conservation is applied for a control volume located immediately downstream of a turbine rotor, while an explicit formula is derived to predict the wake velocity field by using only two parameters as input, namely the

rotor thrust coefficient, $C_t$, and the wake expansion coefficient, $k$. The expression for this wake model is:

$$U_w = U_\infty \left[ 1 - \left( 1 - \sqrt{1-C_t} \right) \left( \frac{D}{D+2k\,x} \right)^2 \right], \tag{6}$$

where $U_w$ is the wake velocity only function of the downstream location, $x$. Known as top-hat wake model, at each downstream location the velocity is uniform within the wake region, which is identified through the wake diameter, $D_w$, which grows linearly in the downstream direction as follows:

$$D_w = D + 2k\,x. \tag{7}$$

The wake expansion coefficient, $k$, is defined in analogy with the jet spreading within shear flows (Pope, 2000). According to the Wind Atlas Analysis and Application Program (WAsP (Mortensen et al., 2011)), the value of the wake expansion coefficient, $k$, is suggested to be equal to 0.075 and 0.05 for onshore and offshore wind farms, respectively. However, in Barthelmie et al. (2010) a lower $k$-value of 0.03 is found to achieve a better agreement with the SCADA data collected for the Nysted offshore

wind farm. The reason for a lower $k$-value might be due to the high occurrence of stable atmospheric conditions for that offshore wind farm.

In Frandsen et al. (2006), a semi-empirical formula is proposed to estimate $k$ by using the aerodynamic roughness length and friction velocity as input. In Peña et al. (2014), a generalized expression for $k$ is proposed, which includes modulations due to the atmospheric stability. In Peña et al. (2016), the same authors provide an empirical formula to predict the wake expansion

factor based on the incoming wind turbulence intensity:

$$k = 0.4 * TI. \tag{8}$$

For the optimization of the parameters of the Jensen model, the thrust coefficient, $C_t$, is varied between 0.01 and 1 with a step of 0.01, while the wake expansion coefficient, $k$, is varied between 0.001 and 0.3 with a step of 0.001. The optimally-tuned wake expansion coefficient, $k_{opt}$, for the Jensen model is reported in Fig. 2(a) against the wind turbulence intensity measured

by the SCADA, $TI$. It is evident that $k_{opt}$ is proportional to $TI$. Furthermore, it is also observed that operative conditions belonging to region 3 of the power curve, namely for incoming wind speed at hub height normalized by the wind turbine rated



wind speed, $U_{hub}^*$, higher than 0.9, are characterized by a wake expansion coefficient lower than 0.04. In contrast, for operative conditions in region 2 of the power curve, $k_{opt}$ grows rapidly with the incoming turbulence intensity approaching values close to 0.1.

In analogy with the model proposed in Peña et al. (2016) (Eq. 8), a linear fitting between $k_{opt}$ and $TI$ is calculated producing a Pearson correlation coefficient of 0.92, intercept of -0.01 and a slope of 0.48, while the slope proposed in Peña et al. (2016) is 0.4. Therefore, this work would suggest a slightly revised model for estimating the wake expansion coefficient for the Jensen wake model as:

$$k_{opt} = 0.48 * TI - 0.01. \tag{9}$$

The optimization of the parameters for the Jensen model also produces estimates of the rotor thrust coefficient, $C_{t_{opt}}$, for the various clusters of the used LiDAR dataset. Fig.2(b) shows that a roughly constant $C_{t_{opt}}$ is observed for the region 2 of the power curve, while approaching a non-dimensional hub-height velocity, $U_{hub}^*$, of about 0.9, a reduction of $C_{t_{opt}}$ is observed as a consequence of the blade pitching operated by the turbine controller to keep power capture equal to the rated power. The rotor thrust coefficients calculated through the mass and streamwise-momentum budgets (Eqs. 2 and 3), $C_{t_{LiDAR}}$, is compared
with that obtained from the optimal tuning of the Jensen model, $C_{t_{opt}}$, in Fig.3(a) for each cluster of the LiDAR dataset. The Pearson correlation coefficient between these two parameters is 0.95, which corroborates the accuracy of the optimal tuning of the Jensen model from the LiDAR data. The comparison of $C_{t_{SCA}}$ with $C_{t_{opt}}$ in Fig.3(d) confirms the previous results presented in Iungo et al. (2018b), namely the thrust coefficient estimated from the SCADA is generally an under-estimation of its actual value because not including drag components not related to the torque generation, namely drag connected with airfoil
stall or bluff-body behavior due to the wind turbine tower and nacelle.

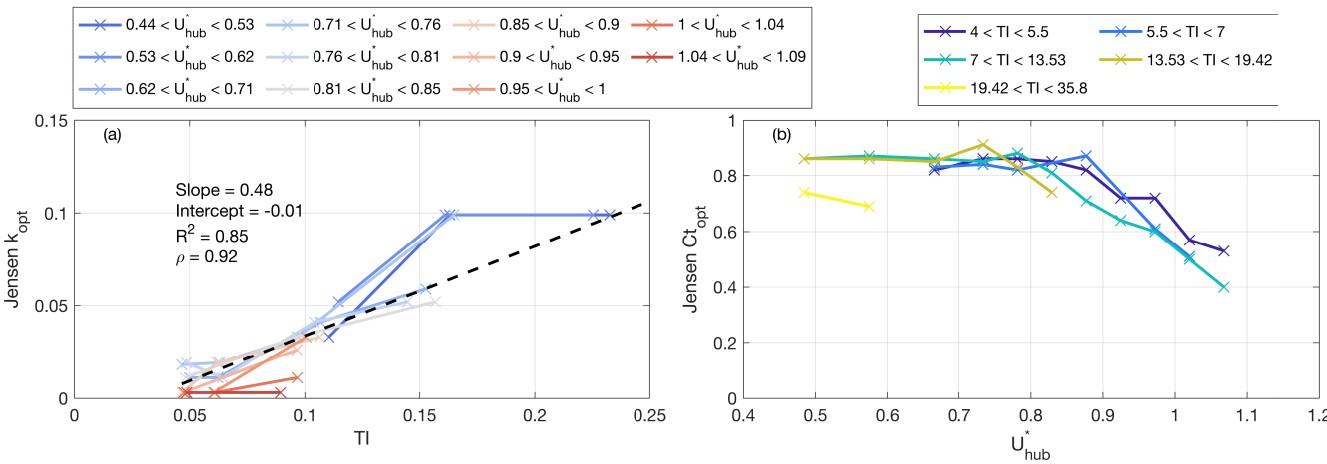

**Figure 2.** Optimal tuning of the Jensen wake model: (a) optimized wake expansion coefficient, $k_{opt}$, as a function of the incoming turbulence intensity, $TI$, and colored by the normalized incoming wind speed at hub height, $U_{hub}^*$. Black dashed line is the linear fitting; (b) optimized thrust coefficient, $C_{t_{opt}}$, as a function of $U_{hub}^*$ and colored according to $TI$.

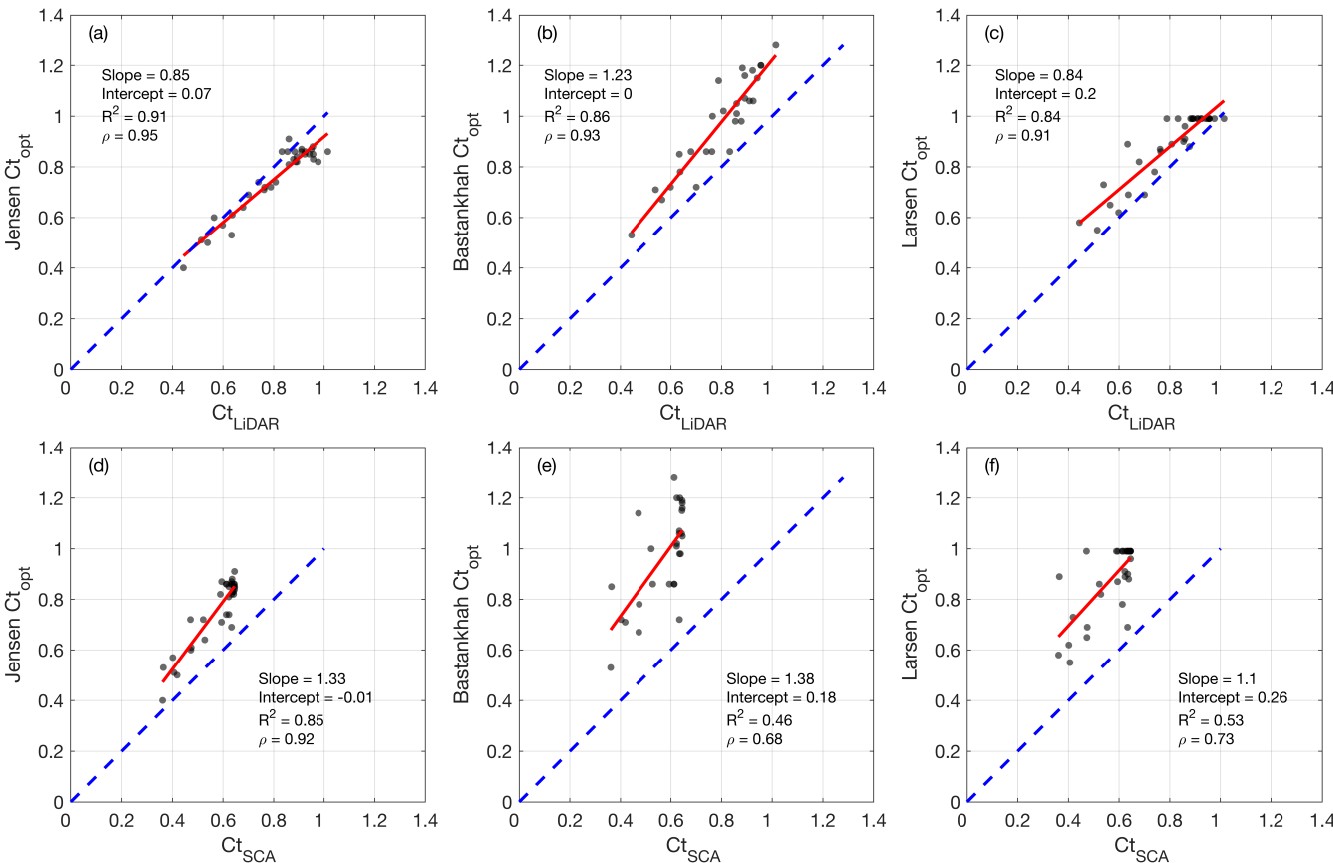

**Figure 3.** Linear regression of the thrust coefficient obtained from the optimal tuning of the wake models, $C_{t_{opt}}$, against that calculated directly from the LiDAR data, $C_{t_{LiDAR}}$ [(a), (b) and (c)], or SCADA data, $C_{t_{SCA}}$ [(d), (e) and (f)]: (a) and (d) Jensen model; (b) and (e) Bastankhah model; (c) and (f) Larsen model. The red solid line is the linear fitting result and the blue dashed line is the 1:1 line.

### 3.2 Bastankhah wake model

For the Bastankhah wake model, a Gaussian profile is used to describe the wake velocity field in the transverse direction at a given downstream location. This Gaussian velocity profile is then used to solve the mass and momentum budgets as for jets evolving in a boundary layer (Tennekes et al., 1972; Bastankhah et al., 2014). The derived self-similar wake velocity profile

can be formulated as:

$$\frac{\Delta U}{U_\infty} = C(x)e^{\left(-\frac{r^2}{2\sigma^2}\right)}, \tag{10}$$

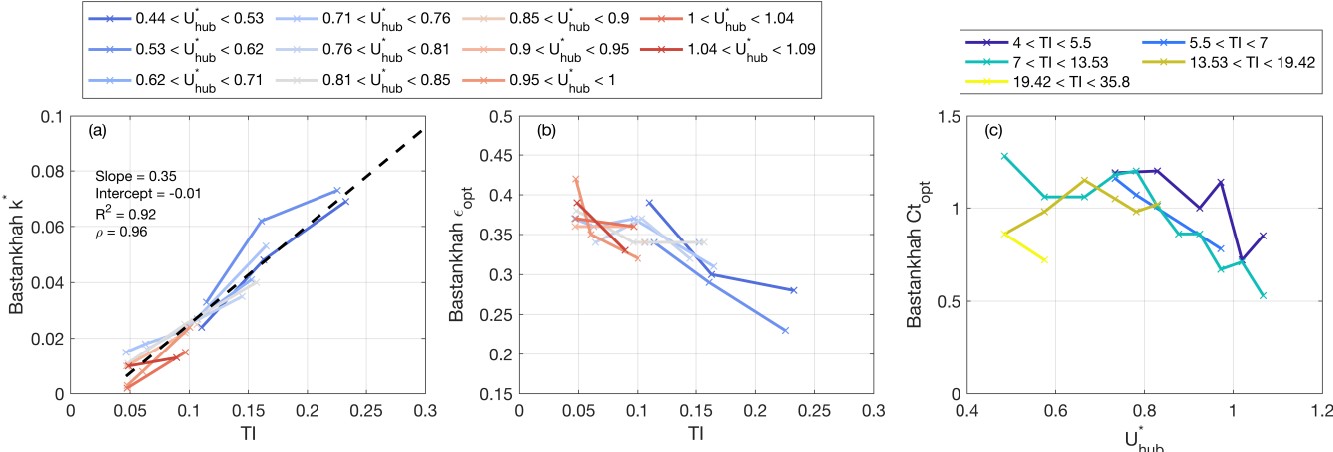

**Figure 4.** Optimally-tuned parameters from Bastankhah wake model: (a) wake expansion parameter, $k^*$, as a function of the incoming wind turbulence intensity, $TI$ (black dashed line is the linear fitting), colored by $U_{hub}^*$; (b) offset of the standard deviation of the Gaussian wake velocity profile, $\epsilon$, as a function of the incoming wind turbulence intensity, $TI$, colored by $U_{hub}^*$; (c) $C_t$ as a function of $U_{hub}^*$, colored by $TI$.

where $\Delta U$ is the wake velocity deficit at the downstream location $x$, $\sigma$ is the standard deviation of the Gaussian velocity profile and $C(x)$ is the maximum velocity deficit. Inheriting a linear wake expansion from the Jensen model, $\sigma$ is modeled as:

$$\frac{\sigma}{D} = k^* \frac{x}{D} + \varepsilon, \tag{11}$$

where $k^*$ is the growth rate of the Gaussian standard deviation and $\varepsilon$ is its offset at the rotor location. The wake velocity field predicted through the Bastankhah wake model can be written as:

$$\frac{\Delta U}{U_\infty} = \left[1 - \sqrt{1 - \frac{C_t}{8(k^*x/D + \varepsilon)^2}}\right] \times exp\left\{-\frac{1}{2(k^*x/D + \varepsilon)^2}\left[\left(\frac{z - z_h}{D}\right)^2 + \left(\frac{y}{D}\right)^2\right]\right\}, \tag{12}$$

where y and z are transverse and vertical coordinates, respectively, while $z_h$ is hub height. For the Bastankhah wake model, the thrust coefficient, $C_t$, is varied between 0.01 and 1.71 with a step of 0.01, while the wake expansion coefficient, $k^*$, is varied

between 0.001 and 0.3 with a step of 0.001. Furthermore, the parameter $\epsilon$ in Eq. 11 is varied between 0.2 and 0.5 with a step of 0.01.

The optimized wake expansion factor of the Bastankhah wake model, $k_{opt}^*$, is reported in Fig. 4(a) as a function of the incoming wind turbulence intensity. In agreement with the results obtained for the wake expansion factor of the Jensen model, $k_{opt}^*$ also increases monotonically with the incoming turbulence intensity. Furthermore, a secondary trend with the rotor thrust

coefficient is observed. Indeed, the operative conditions with $U_{hub}^* > 0.9$ are characterized by a slightly smaller $k_{opt}^*$. Similarly to a previous work (Carbajo et al., 2018), the linear fitting bend $TI$ is calculated producing the following optimal values: $k_{opt}^* = 0.34 * TI - 0.013$, with R-square value of 0.96. The slope between $k_{opt}^*$ and $TI$ of 0.34 is equal to that found in Carbajo et al. (2018).





In Fig.4(b), the offset of the standard deviation of the velocity profile, $\epsilon$ (Eq.11), decreases with reducing $TI$. This is in
agreement with the faster mixing and recovery of the wake, which leads to a shorter near-wake region. Even a secondary trend
of $\epsilon$ is detected as a function of $U_{hub}^*$. These results suggest that the lower $C_t$ value associated with high $U_{hub}^*$, leads to a
narrower and shallower velocity deficit in the near wake region than for operations in region 2 of the power curve.

Fig. 4(c) shows the optimized thrust coefficient, $C_{t_{opt}}$, against the normalized hub-height velocity, $U_{hub}^*$. Similarly to the
results obtained for the Jensen wake model, the $C_{t_{opt}}$ is practically uniform in region 2 of the power curve ($U_{hub}^* < 0.9$), then it
monotonically decreases in region 3 of the power curve ($U_{hub}^* > 0.9$). The thrust coefficient obtained from the optimal tuning
of the Bastankhah wake model is then compared with that obtained from the mass and momentum budgets applied to the
LiDAR data in Fig.3(b), and the respective values derived from the SCADA data in Fig.3(e). It is noteworthy that $C_{t_{opt}}$ can be
larger than 1, as a result of Eq. 12 for which a real solution can only be obtained for $x/D \geq (\sqrt{C_t/8} - \epsilon)/k^*$. This constraint
has been added for the optimization of the parameters of the Bastankhah wake model, which results in rejecting some LiDAR
data in the near wake. Furthermore, removing LiDAR data collected in the near wake can be beneficial for the optimal tuning
of the Bastankhah model, because in the near wake the velocity profile can be significantly different from the typical Gaussian
shape, which is an underlying assumption for this wake model.

Similarly to Carbajo et al. (2018), the detection of the near- to far-wake transition is associated with the downstream location
where the fitting of the streamwise velocity as a function of the radial position with a Gaussian function produces a Pearson
correlation coefficient larger than 0.99. In Fig. 5(a), the velocity profiles for the cluster with $U_{hub}^* \in [0.71, 0.76]$ and $TI \in [7,$
13.5] % are reported. Based on the before-mentioned criterion, the near wake region ends at $x = 1.75D$. However, a difference
of 0.05 in the minimum of the normalized velocity is observed between the measured and fitted profile. As a consequence,
the Bastankhah wake model overestimates the maximum velocity deficit to maximize the correlation between the data and
the Gaussian fitting, especially in proximity of the sides of the wake. The drawback of this fitting procedure consists in an
over-estimation of $C_t$. Rejecting LiDAR data in the near-wake region for the optimization of the e Bastankhah wake model
can be, thus, beneficial for a more accurate estimation of $C_t$. Fig 5(b) shows that the percentage error, $PE$, obtained by using
LiDAR data from the entire wake region is larger than only using far-wake LiDAR data. On average, the error drops down by
15.8% from the full wake cases, while the maximum improvement is of 69.5%.

### 3.3   Larsen model

For the Larsen wake model, the RANS equations are simplified by neglecting gradients with a smaller order of magnitude
considering the boundary layer approximation and dropping the viscous term due to the high Reynolds numbers involved
for applications to utility-scale wind turbines (Swain, 1929). The axial velocity field is solved by leveraging the similarity
solution and using the mixing length model as turbulence closure. The first-order contribution to the axial velocity prediction
is expressed as:

$$\Delta U_1 = -\frac{1}{9}\left(C_t\,\frac{A}{(x+x_0)^2}\right)^{1/3}\left\{r^{-3/2}\left[3c_1^2 C_t\,A(x+x_0)\right]^{-1/2} - \left(\frac{35}{2\pi}\right)^{3/10}(3c_1^2)^{-1/5}\right\}^2, \qquad (13)$$



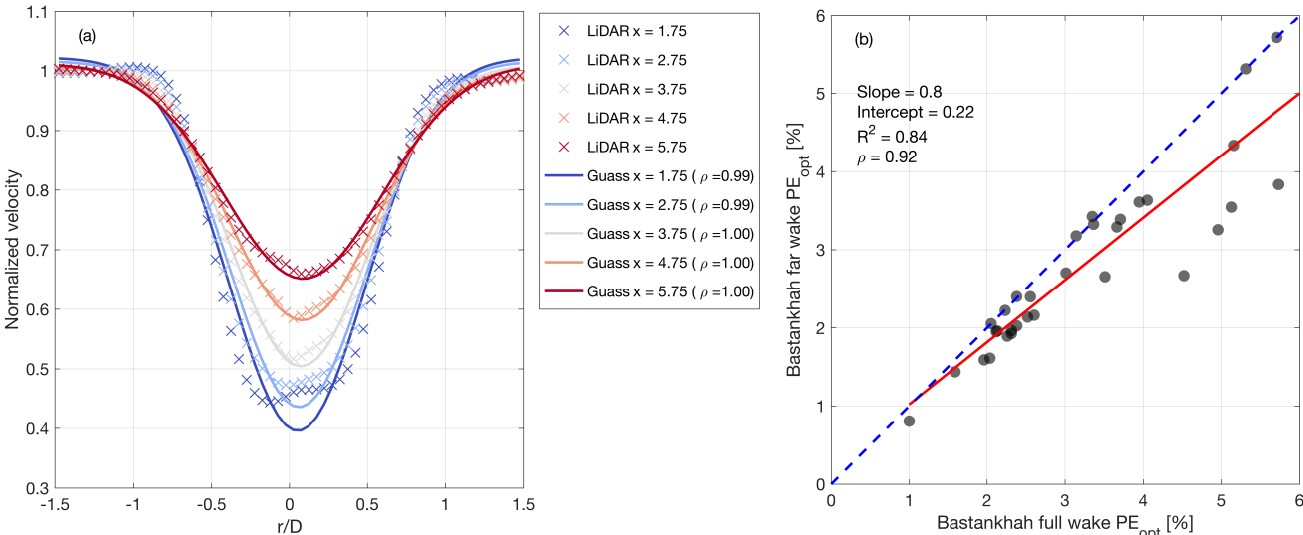

**Figure 5.** (a) Normalized velocity profiles from a LiDAR cluster with $U_{hub}^* \in [0.71, 0.76]$ and $TI \in [7, 13.5]$ %. The cross marker represents LiDAR measurements, solid lines indicate Gaussian fitting, and different colors indicate downstream location; (b) linear regression between optimized average $PE$ retrieved by using the LiDAR data for the entire wake or excluding the near-wake region. The red solid line is the linear fit and the blue dashed line represents 1:1 ratio.

where $\Delta U_1$ is the wake velocity deficit, $C_t$ is the thrust coefficient of the turbine rotor, $x_0$ is the streamwise offset for the reference frame, $A$ is the rotor area and $c_1$ is a constant related to mixing length model. The wake region is identified through the following wake radius:

$$R_w = \left(\frac{35}{2\pi}\right)^{1/5} (3c_1^2)^{1/5} \left[C_t \, A(x + x_0)\right]^{1/3}. \tag{14}$$

The coefficient $c_1$, $x_0$ are calculated by following the calibration procedure in (Larsen et al., 2003). For more details, the reader can refer to the Appendix A. The radial velocity for the Larsen wake model is calculated as:

$$u_r = \frac{1}{3}(C_t A(x))^{1/3} x^{-3/5} r \left\{ r^{-3/2}(3c_1^2 C_t A(x + x_0))^{-1/2} - \left(\frac{35}{2\pi}\right)^{3/10}(3c_1^2)^{-1/5} \right\}^2. \tag{15}$$

To satisfy the continuity constraint, the coefficient 1/3 should be changed to -1/27; for more details see the Appendix B. In Larsen, G. C. (2009), a formula for the second-order contribution to the axial velocity field is provided as:

$$\Delta U_2 = \left(C_t \, \frac{A}{(x + x_0)^2}\right)^{2/3} \sum_{i=0}^{4} d_i z(x, r)^i. \tag{16}$$

For the Larsen model, both first and second-order contributions require two fundamental input parameters: the thrust coefficient, $C_t$, and the incoming wind turbulence intensity, $TI$. $c_1$ is calibrated through $x_0$ and $C_t$ (Eq. A1). However, we seek for a more data-driven approach to compute the velocity field with the Larsen wake model, yet avoiding the proposed empirical formulas





**Figure 6.** (a) Percentage error, $PE$, between the Larsen-model predictions with or without second-order solution; the color bar indicates incoming turbulence intensity of the considered LiDAR cluster. Color maps of the wake velocity field for the LiDAR cluster with $0.85 < U_{hub}^* < 0.90$ and $4\% < TI < 5.5\%$: (b) LiDAR data; (c) and (d) are prediction through the Larsen model without or with, respectively, second-order contribution (green lines are the location of wake edge); (e) and (f) percentage error with respect to the LiDAR data without or with, respectively, second-order contributions.

for $x_0$ and $c_1$. Therefore, in this work, we consider $C_t$, $c_1$ and $x_0$ as input parameters, whose physical interpretation will be

further illustrated in the following.

 In the case study shown in Fig. 6b, it is presented the ensemble-averaged wake velocity field for the LiDAR cluster with hub-height velocity range of $0.71 < U_{hub}^* < 0.76$ and turbulence intensity range of $4 < TI < 5.5\%$. The optimally-tuned velocity field obtained with or without the second-order contribution of the Larsen model are reported in Fig. 6(d) and 6(c), respectively. The first-order solution of the Larsen model seems to predict higher wind speed at the wake edge and a lower velocity in

proximity to the wake center. In contrast, the second-order solution seems to be more accurate and characterized by a lower





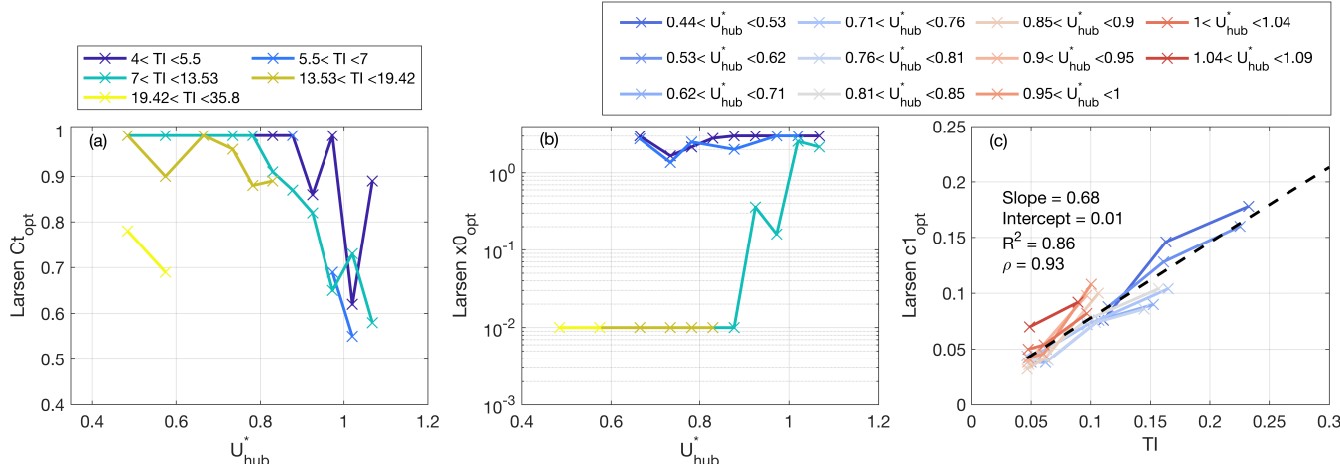

**Figure 7.** Optimally-tuned parameters for the Larsen wake model: (a) $C_t$ and (b) $x_0$ as a function of the hub-height velocity, $U^*_{hub}$, for each LiDAR cluster, while the lines are colored by the incoming turbulence intensity (legend at the top left corner); (c) Prandtl's mixing-length parameter, $c_1$, versus incoming turbulence intensity and colored by $U^*_{hub}$ (legend at the top right corner). The black dashed line is the linear fitting between $c_1$ and TI.

$PE$. Therefore, we used the second-order solution for the optimal tuning of the Larsen wake model and found the relative norm of $PE$ decreases by -21.4 % on average for all the LiDAR clusters compared to the case with only considering the first-order solution (Fig. 6(a)).

The optimally-tuned parameters for the Larsen wake model are reported in Fig. 7. The model has successfully captured the

reduction of the thrust coefficient for operations in region 3 of the power curve, regardless of the incoming turbulence intensity, $TI$. In Fig. 7(b), the parameter $x_0$ converges to 0 when the incoming turbulence intensity increases. As mentioned in section 3.3, $x_0$ is defined as the distance between the rotor position and the origin of the used coordinate system. Nonetheless, it can also be denoted physically as the position where the initial wake width equals to one rotor diameter. Therefore, a faster wake recovery rate due to higher incoming turbulence makes this condition occurring closer to the turbine rotor.

Regarding the wake recovery rate reported in Fig. 7(c), we can see the enhancement of turbulent mixing as a function of increasing turbulence intensity, which can be modeled through a linear function with a slope of 0.68 and interception of 0.01. However, the secondary effect on turbulent diffusion due to $C_t$ is not singled out for the Larsen wake model. The justification can be found in the calibration procedure. Assuming an increase of $C_t$, the effective rotor $D_{eff}$ increases monotonically. If we substitute Eq. A2 into Eq. A1 and recast it, we obtain:

$$c_1 \sim C_t^{-\frac{5}{6}} \left( \frac{9.5}{(2R_{9.5})^3 - D_{eff}^3} \right)^{-\frac{5}{6}}, \tag{17}$$

which suggests that $c_1$ automatically decreases.





### 3.4 Ainslie model

Similarly to the Larsen wake model, the Ainslie wake model is derived from the RANS equations for incompressible flows
(Ainslie, 1988). The turbulent eddy viscosity (EV) is formulated as follows:

$$EV = F[k_l b(U_\infty - U_c) + K_M], \tag{18}$$


where $k_l$ is a constant expected to be a function of the wake shear rather than incoming turbulence intensity. In Ainslie (1988),
a suggested value of 0.015 is proposed, which was obtained from wind tunnel experiments. The parameters $b$ and $U_\infty - U_c$ are
the wake width and velocity deficit, respectively. $K_M$ is the eddy diffusivity for momentum, which is defined as:

$$K_M = (\kappa u^* z)/\phi_m(z/L), \tag{19}$$


where $\kappa$ is the Von Kármán constant and $u^*$ is the friction velocity. $\phi_m$ and $z/L$ are two dimensionless groups that define
the Businger-Dyer relationships (Stull, 1988). Furthermore, a filter function, $F$, is introduced to model effects of the wake-
generated turbulence:

$$F(x) = \begin{cases} 0.65 + \left[\frac{x-4.5}{23.32}\right]^{1/3} & x \le 5.5 \\ 1 & x > 5.5 \end{cases} \tag{20}$$

It is noteworthy that the filter function, $F$, wake width, $b$, and velocity deficit, $U_0 - U_c$, are all functions of the downstream

distance from the rotor. Therefore, the turbulent eddy viscosity, $EV$, is also a function of the downstream position and coupled
with the solution of the wake velocity field. For the sake of reducing the required computational costs, the equations of the
Ainslie wake model are solved with a parabolic approach advancing in the downstream direction.

For the Ainslie wake model, the wake width, $b$, is defined as the radial location where the wake velocity is equal to 90% of
the freestream velocity. Similarly to Kim et al. (2018), we adopted as initial wake velocity profile the experimental LiDAR data

measured at a downstream distance of 1.25 D. In this regard, the Ainslie model provides the advantage of using experimental
data as initial wake velocity profile, as long as the data are axisymmetric per the model formulation. For more details on the
numerical solution of the Ainslie wake model see Appendix C.

Summarizing, the inputs of the Ainslie model are the thrust coefficient, $C_t$, turbulence intensity, $TI$, filter function, $F$, shear
layer constant, $k_l$, and eddy diffusivity of momentum, $K_M$. In this study, we set the filter function, $F$ (Eq. 20) equal to 1

throughout the whole wake region for the sake of simplicity. The wake-generated turbulence is taken into account through
the parameters $k_l$ and $K_M$. It is noteworthy that $C_t$ and $TI$ are only used to tune the initial wake profile at the downstream
location $x = 1.25D$, where the LiDAR measurements are available. Therefore, the independent parameters required for the
optimal tuning of the Ainslie wake model are $k_l$, which is varied between 0.001 and 0.101 with a step of 0.005, and $K_M$,
which is varied between 0.001 and 0.501 with a step of 0.002.

Since the two model parameters $k_l$ and $K_M$ are directly hinged on the eddy viscosity and, thus, with turbulence mixing
and wake recovery rate, in Fig. 8 we show them as a function of the incoming turbulence intensity. The parameter $k_l$ is the



weighting that measures the contribution from wake deficit and wake width to the eddy viscosity. In Fig. 8(a) it is interesting to note that $k_l$ seems to be independent of hub-height velocity. It has a peak at the $TI$ of about 7%, then it reduces to zero when the incoming turbulence intensity exceeds about 15%. In contrast, the parameter $K_{M_{opt}}$ is proportional to the incoming

wind turbulence intensity, as shown in Fig. 8(b). A similar trend is obtained for the eddy viscosity in Fig. 8(c). Furthermore, we calculated the standard deviation of eddy viscosity along the $x$-direction for all the LiDAR clusters (not shown here) and found that it is two orders of magnitude smaller than the average eddy viscosity for the respective LiDAR cluster. This suggests that a constant eddy viscosity model can well reproduce the downstream evolution of the wake velocity field. Finally, these results suggest that the turbulent eddy viscosity can be modeled as: $EV_{opt} = 0.14 * TI - 0.01$ with a Pearson correlation coefficient

of 0.95.

## 4    Results and discussion

Once the parameters of the four considered engineering wake models have been optimally tuned based on the mean velocity fields retrieved from the LiDAR measurements, and their trends as functions of the normalized incoming wind speed at hub height, $U^*_{hub}$, and turbulence intensity, $TI$, have been discussed, it is worth to scrutinize more in detail the predictions generated

from the wake models. For instance, the mean velocity field measured from the LiDAR for the cluster with $U^*_{hub} \in [0.76, 0.81]$ and $TI \in [13.5\%, 19.4\%]$ is reported in Fig. 9 and compared with the respective predictions obtained from the selected wake models. Firstly, the simplistic predictions obtained through the Jensen model are evident, even though overall information on the mean kinetic energy content available within the wake and its evolution in the downstream direction are provided.

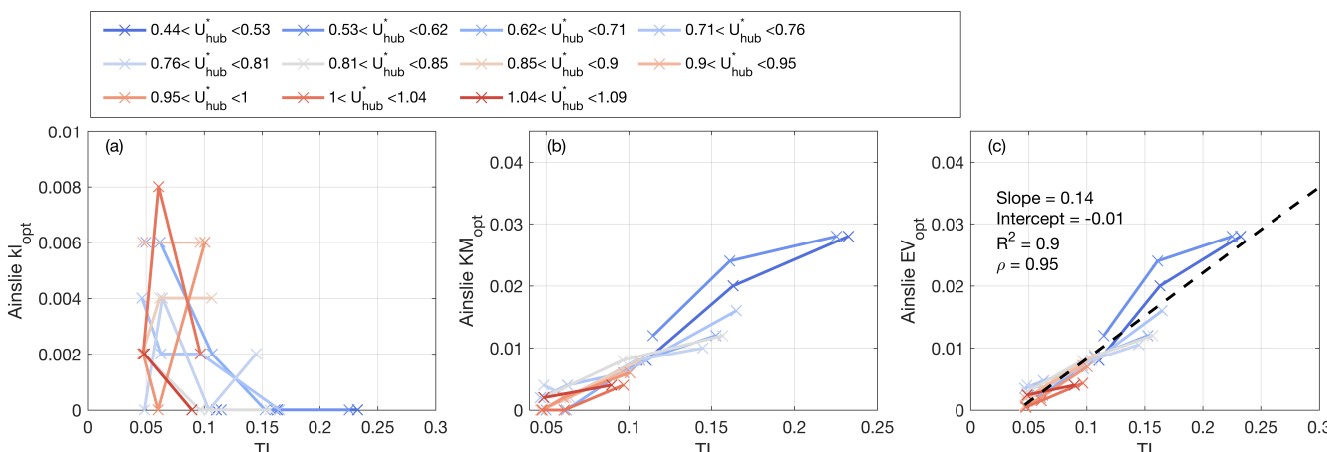

**Figure 8.** Optimal tuning of the parameters for the Ainslie wake model: (a) $kl$, (b) $KM$ and (c) eddy viscosity versus the incoming turbulence intensity of each LiDAR cluster, while the lines are colored by the hub-height velocity. The black dashed line represents linear fitting.





The velocity field predicted through the Bastankhah wake model looks very similar to the mean velocity field measured by
the LiDAR, especially in the far wake, indicating that the velocity profiles in the radial direction can be modeled with a good
level of accuracy through a Gaussian function, which is the underlying assumption of the Bastankhah wake model. A larger
wake velocity deficit with respect to the reference LiDAR data is observed in the near wake for the model predictions. This
feature of the Bastankhah wake model can be better understood through the velocity profiles in the radial direction reported
for various downstream locations and incoming turbulence intensity in Fig. 10. For these data clusters, which are calculated
for incoming wind speed within the range $0.62 < U^*_{hub} < 0.71$ and different $TI$, it is observed that in the near wake the mean
velocity field measured by the LiDAR is not axisymmetric and, more importantly, it is significantly different from a Gaussian
function (Zhan, et al., 2019). The velocity profiles recover a more Gaussian-like trend by moving downstream and/or increasing
the incoming turbulence intensity, $TI$. The optimization procedure of the Bastankhah wake model attempts to maximize the
agreement between the model predictions and the LiDAR data, especially in proximity of the sides of the wake, by enhancing
the maximum velocity deficit, which often results in an over-estimated thrust coefficient (see Fig. 5(a)). As discussed in section
3.2, this feature has motivated the rejection of wake regions for the fitting of the LiDAR data with a Gaussian function with
Pearson correlation coefficient smaller than 0.99 for the optimal tuning of the model.

Predictions of the near-wake velocity field are improved for the Larsen wake model (Fig. 9(d)). Furthermore, a very good
accuracy is generally observed throughout the downstream evolution of the wake, which suggests that the use of the RANS
equations with the mixing-length turbulence closure model is an efficient strategy to predict accurately wind turbine wake, yet
with very low computational costs. Compared to the empirical modeling of wake expansion in Jensen and Bastankhah wake

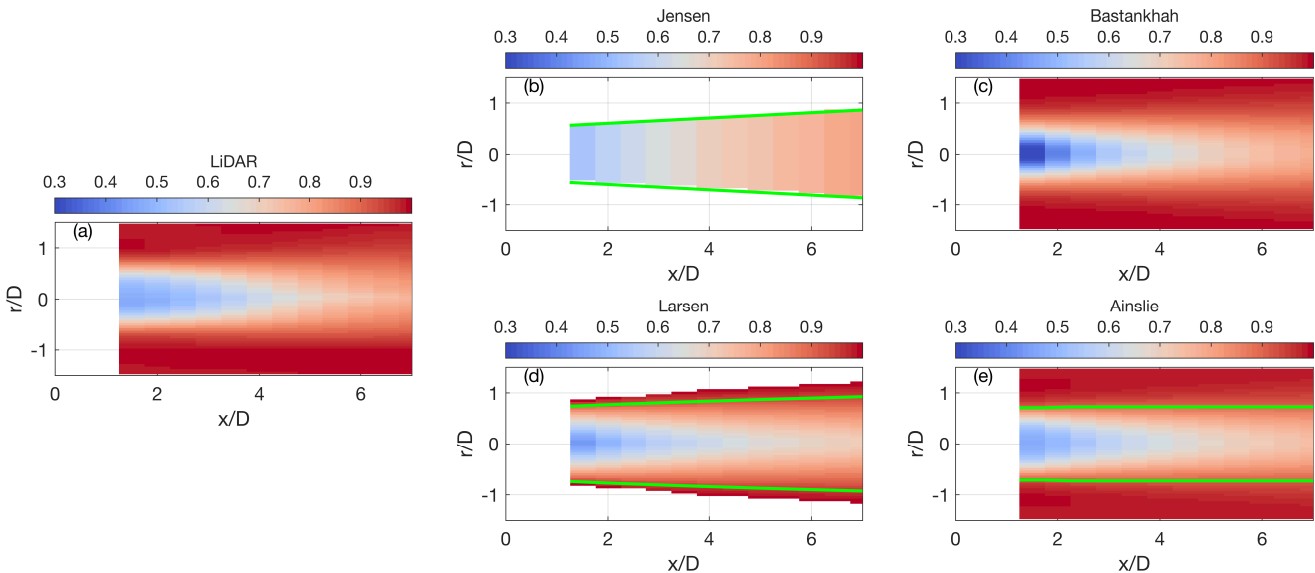

**Figure 9.** Cluster with $U^*_{hub}$ of [0.76, 0.81] and $TI$ of [13.5%, 19.4%] : (a) LiDAR data; (b) Jensen wake model; (c) Bastankhah wake model;
(d) Larsen wake model; (d) Ainslie wake model.





models, the mixing-length approach not only provides more clarity in interpreting the role of turbulence but also defines the wake width without ambiguity (green line in Fig 9). Prediction accuracy is further enhanced with the Ainslie wake model, where a modeling strategy similar to that of the Larsen model is used with the undeniable advantage of using the mean velocity

field measured through the LiDAR at $x = 1.25\,D$ as the initial condition at the near wake. For general applications, information about the wake velocity field at the near wake might be not available and, thus, this input should be replaced by a modeling approach, or in alternative by previous experimental or numerical datasets.

**Figure 10.** Velocity profiles predicted from four wake models and compared with LiDAR data for $0.62 < U^*_{hub} < 0.71$ and different values of $TI$: first row, $4\% < TI < 5.5\%$; second row, $5.5\% < TI < 7\%$; third row, $7\% < TI < 13.5\%$; fourth row, $13.5\% < TI < 19.4\%$, and different downstream location: first column, at $X = 1.75D$; second column, at $X = 3.75D$; third column, at $X = 5.75D$.



Accuracy in the wake-flow predictions obtained through the optimally-tuned engineering wake models is quantified through the average percentage error, $PE$ (Eq. 1), which is reported in Fig. 11 for the various clusters of the LiDAR dataset. It is

evident that the Jensen model is characterized by the lowest, yet comparable, accuracy, which is a consequence of the top-hat representation of the wake velocity field. The Larsen model has a better accuracy than the Jensen model, but worse than Ainslie and Bastankhah models. The Bastankah model produces in general similar percentage error as Ainslie model for different LiDAR clusters. It is noteworthy that the accuracy in model prediction generally increases for higher incoming turbulence intensity, $TI$. Indeed, the enhanced turbulent mixing leads to smoother and more Gaussian-like wake velocity profiles.

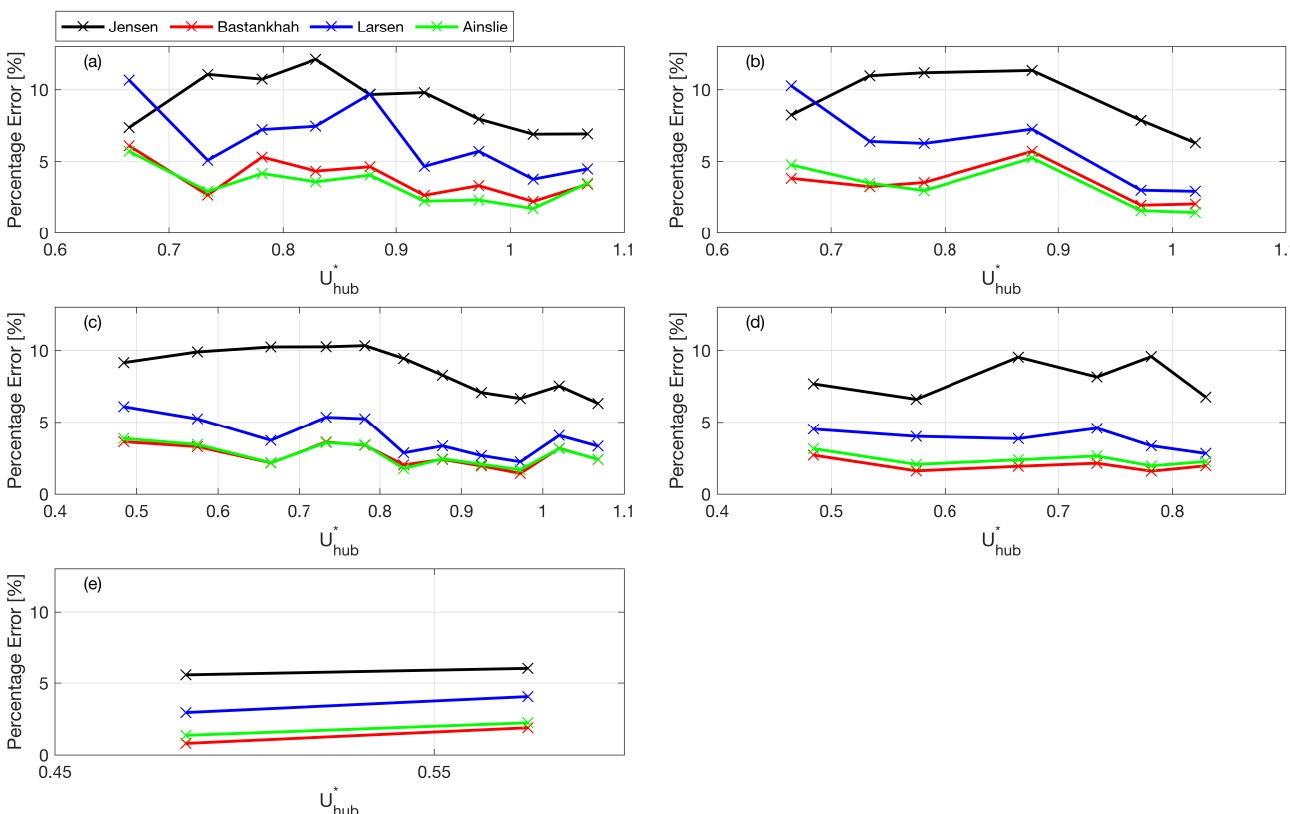

**Figure 11.** Percentage error, $PE$ (Eq. 1) for four engineering wake models and various LiDAR clusters based on incoming turbulence intensity, $TI$: (a) $4\% < TI < 5.5\%$; (b) $5.5\% < TI < 7.0\%$;(c) $7.0\% < TI < 13.5\%$;(d) $13.5\% < TI < 19.4\%$;(e) $19.4\% < TI < 35.8\%$.

Finally, a sensitivity analysis of the optimally-tuned model parameters as a function of $C_t$ and $TI$ is provided. To this aim, both input and output parameters are normalized as follows:

$$\hat{f} = \frac{f - f_{min}}{f_{max} - f_{min}}, \tag{21}$$





**Table 1.** Linear regression between inputs from LiDAR cluster and output parameters from the wake models.

| fine-tuned parameters | Jensen model | | Bastankhah model | | | Larsen model | | | Ainslie model | |
|---|---|---|---|---|---|---|---|---|---|---|
| | $\widehat{C_t}$ | $\widehat{k}$ | $\widehat{C_t}$ | $\widehat{k^*}$ | $\widehat{\epsilon}$ | $\widehat{C_t}$ | $\widehat{c_1}$ | $\widehat{x_0}$ | $\widehat{k_l}$ | $\widehat{K_M}$ |
| Slope($\widehat{C_{t\,LiDAR}}$) | **0.95** | 0.24 | **0.93** | 0.12 | 0.24 | **1.1** | -0.23 | -0.43 | -0.19 | -0.06 |
| Intercept($\widehat{C_{t\,LiDAR}}$) | **0.09** | 0.16 | **-0.01** | 0.24 | 0.35 | **0.07** | 0.45 | 0.71 | 0.39 | 0.23 |
| $R^2(\widehat{C_{t\,LiDAR}})$ | **0.91** | 0.04 | **0.86** | 0.01 | 0.11 | **0.84** | 0.05 | 0.07 | 0.03 | 0 |
| $\rho(\widehat{C_{t\,LiDAR}})$ | **0.95** | 0.19 | **0.93** | 0.12 | 0.33 | **0.91** | -0.23 | -0.26 | -0.16 | -0.05 |
| Slope($\widehat{TI}$) | 0.14 | **1.12** | -0.14 | **0.92** | -0.55 | -0.13 | **0.86** | -1.27 | -0.54 | **0.97** |
| Intercept($\widehat{TI}$) | 0.66 | **0.01** | 0.63 | **0.06** | 0.66 | 0.80 | **0.07** | 0.78 | 0.42 | **-0.01** |
| $R^2(\widehat{TI})$ | 0.02 | **0.85** | 0.02 | **0.93** | 0.61 | 0.01 | **0.86** | 0.60 | 0.25 | **0.91** |
| $\rho(\widehat{TI})$ | 0.15 | **0.92** | -0.15 | **0.96** | -0.78 | -0.11 | **0.93** | -0.78 | -0.5 | **0.95** |

where $f$ is a generic input or output parameter. Through the linear regression of the normalized parameters, we report four quantities, namely slope, intercept, R-square value and Pearson correlation coefficient $\rho$. In Table 1, the numbers in bold highlight the dominant correlations between model parameters and input parameters, i.e. $C_t$ and $TI$.

As mentioned above, the optimal tuning of the Jensen, Bastankhah and Larsen wake models generates an estimate of the thrust coefficient of the turbine rotor, $C_t$. The linear regression of the optimally-tuned $C_t$ with the respective values obtained from the LiDAR data through the application of mass and momentum budgets (section 3), $C_{t_{LiDAR}}$, produces a slope between 0.93 and 1.1. This indicates a very good accuracy of the optimal tuning procedure, especially considering that for the linear regression the $R$-square value is always larger than 0.84 and the Pearson correlation coefficient larger than 0.91.

Regarding the model parameters representing the wake turbulent diffusion, the slope obtained through the linear regression of these parameters with the incoming turbulence intensity, $TI$, is between 0.86 and 1.12 ($R$-square and Pearson correlation coefficient larger than 0.85 and 0.92, respectively). These results further corroborate the models already proposed by Peña et al. (2016); Carbajo et al. (2018), for which the wake turbulent expansion is always assumed linearly proportional to $TI$.

## 5 Conclusions

Low-computational costs and easy implementation are key factors for the wide application of engineering wake models in wind energy, for both industrial and academic pursuits. However, it is challenging to tune parameters of these engineering wake models to achieve a satisfactory accuracy for predictions of wakes and power capture required for design and control of wind farms. Furthermore, this calibration can be even more challenging when wake models are used for a broad range of atmospheric stability regimes or in presence of flow distortions induced by the site topography.

In this paper, we have considered four widely-used engineering wake models, namely the Jensen model, Bastankhah model, Larsen model and Ainslie model. The tuning parameters of these engineering wake models have been optimally calibrated





by minimizing the mean percentage error between the wake flow predicted through the models and the mean velocity fields measured through a scanning Doppler wind LiDAR deployed at an onshore wind farm in North Texas. Statistics of the wake

velocity field are obtained through a cluster analysis based on the incoming turbulence intensity at hub height and the normalized hub-height wind speed. The results of the optimal-tuning procedure have shown that the thrust coefficient obtained through this numerical approach is in very good agreement with the values obtained by applying the mass and streamwise momentum budgets on the mean LiDAR data by neglecting pressure gradients and turbulent stresses. Furthermore, the model parameters representing the wake turbulent expansion and recovery are roughly linearly proportional to the incoming wind

turbulence intensity at hub height.

This study has shown that the Jensen model has a lower yet comparable accuracy than the remaining three wake models, which is mainly connected with the simplistic top-hat assumption used for modeling the wake velocity deficit. However, a good estimate of the mean kinetic energy within the wake as a function of the downstream location is achieved through the Jensen wake model. The Larsen wake model has generally shown better accuracy than the Jensen model yet lower than for the

Bastankhah and Ainslie wake models. This feature seems to be the effect of a slightly more complex formulation of the model, leading to the presence of parameters that are not easy to tune through a data-driven approach, as for this work.

The Bastankhah wake model has shown great accuracy in wake predictions upon the optimal tuning of the model parameters for a broad range of incoming turbulence intensity and incoming wind speed at hub height, namely thrust coefficient of the turbine rotor. The main assumption of the Bastankhah wake model consists of modeling the wake velocity profile in the radial

direction through a Gaussian function. Therefore, significant differences between the predictions and the LiDAR data have been observed in the near wake and/or for relatively low incoming turbulence intensity for which the velocity profiles can be not axisymmetric and differ from a Gaussian-like profile. Therefore, we recommend using the Bastankhah wake model only for downstream locations and wind conditions for which the Pearson correlation coefficient between the actual velocity field and the Gaussian model is expected to be higher than 0.99.

Finally, the Ainslie wake model has shown great accuracy indicating that the mixing-length model for the RANS equations is a simple yet efficient turbulence closure model to capture the effects of incoming turbulence and wake-generated turbulence on wake downstream evolution and recovery. The Ainslie wake model provides a great advantage to use as input the velocity profile at a specific streamwise location. This input can be obtained through experiments, numerical simulations or other models.

*Acknowledgements.* This research has been funded by a grant from the National Science Foundation CBET Fluid Dynamics, award number 1705837. This material is based upon work supported by the National Science Foundation under grant IIP-1362022 (Collaborative Research: I/UCRC for Wind Energy, Science, Technology, and Research) and from the WindSTAR I/UCRC Members: Aquanis, Inc., EDP Renewables, Bachmann Electronic Corp., GE Energy, Huntsman, Hexion, Leeward Asset Management, LLC, Pattern Energy, and TPI Composites. Any opinions, findings, and conclusions or recommendations expressed in this material are those of the authors and do not necessarily reflect the

views of the sponsors.



*Data availability.* The used LiDAR dataset is publicly available to download at https://zenodo.org/record/3604444.XiiTdy2ZPUI

*Code and data availability.* The code for the optimal tuning of the models is available on https://www.utdallas.edu/windflux/

**Appendix A: Calibration procedure of first- and second-order solutions of the Larsen wake model**

For the Larsen wake model, the coefficient representing the wake turbulent diffusion is:

$$c_1 = \left(\frac{D_{eff}}{2}\right)^{5/2}\left(\frac{105}{2\pi}\right)^{-1/2}(C_t\,A\,x_0)^{-5/6}, \tag{A1}$$

where the rotor position, $x_0$, is calculated as:

$$x_0 = \frac{9.5D}{\left(\frac{2R_{9.5D}}{D_{eff}}\right)^3}, \tag{A2}$$

while the effective rotor diameter $D_{eff}$ is calculated as:

$$D_{eff} = D\sqrt{\frac{1+\sqrt{1-C_t}}{2\sqrt{1-C_t}}}. \tag{A3}$$

The wake radius at a distance 9.5 rotor diameters downstream, $R_{9.5}$, is calculated as follows:

$$R_{9.5} = 0.5[R_{nb} + min(H, R_{nb})], \tag{A4}$$

with the empirical formula to calculate $R_{nb}$ is:

$$R_{nb} = max\left[1.08D, 1.08D + 21.7D(TI - 0.05)\right], \tag{A5}$$

noting that $TI$ is the incoming wind turbulence intensity at hub height. Eq. A4 includes the blockage effect from the ground as the wake radius could not be larger than hub-height (Larsen et al., 2003; Renkema, 2007). Subsequently, Larsen added another empirical expression for $R_{9.6}$ that consists of input parameters $C_t$ and $TI$, and can be written as:

$$R_{9.6} = a_1 exp(a_2 C_t^2 + a_3 C_t + a_4)(b_1 TI + 1)D, \tag{A6}$$

where all the constants can be found in Larsen, G. C. (2009). The only difference between these two calibration procedures is the calculation of the wake radius at 9.5 $D$ or 9.6 $D$.

For the terms in second-order contribution of Larsen model solution, they are defined as:

$$z(x,r) = r^{3/2}(C_t A(x+x_0))^{-1/2}\left(\frac{35}{2\pi}\right)^{-3/10}(3c_1^2)^{-3/10}, \tag{A7}$$

and

$$d_0 = \frac{4}{81}\left[\left(\frac{35}{2\pi}\right)^{1/5}(3c_1^2)^{-2/15}\right]^6 \times \left(-1-3\left(4-12\left(6+27\left(-4+\frac{48}{40}\right)\frac{1}{19}\right)\frac{1}{4}\right)\frac{1}{5}\right)\frac{1}{8}, \tag{A8}$$






$$d_1 = \frac{4}{81} \left[ \left( \frac{35}{2\pi} \right)^{1/5} (3c_1^2)^{-2/15} \right]^6 \times \left( 4 - 12 \left( 6 + 27 \left( -4 + \frac{48}{40} \right) \frac{1}{19} \right) \frac{1}{4} \right) \frac{1}{5},$$  (A9)

$$d_2 = \frac{4}{81} \left[ \left( \frac{35}{2\pi} \right)^{1/5} (3c_1^2)^{-2/15} \right]^6 \times \left( 6 + 27 \left( -4 + \frac{48}{40} \right) \frac{1}{19} \right) \frac{1}{4},$$  (A10)

$$d_3 = \frac{4}{81} \left[ \left( \frac{35}{2\pi} \right)^{1/5} (3c_1^2)^{-2/15} \right]^6 \times \left( -4 + \frac{48}{40} \right) \frac{1}{19},$$  (A11)


$$d_4 = \frac{4}{81} \left[ \left( \frac{35}{2\pi} \right)^{1/5} (3c_1^2)^{-2/15} \right]^6 \frac{1}{40}.$$  (A12)

**Appendix B: A note on the Larsen wake model**

The authors noticed that for roughly identical predictions in streamwise velocity component from the Larsen and Ainslie wake models, the radial velocity predicted from the former is one order of magnitude larger than that for the latter, while

having the opposite sign. Subsequently, we calculated the divergence in cylindrical coordinate and non-conservative form, $\left( \frac{\partial u_x}{\partial x} + \frac{u_r}{r} + \frac{\partial u_r}{\partial r} \right)$, for both models. For the case in Fig. B1, the input $C_t$ and turbulence intensity are set equal to 0.9 and 12 %, respectively. The $u_x$ profile at $x = x_0$ obtained from the Larsen wake model is used as the initial profile for the Ainslie wake model, while $K_M = 0.01$ and $k_l = 0.015$ are used for turbulence closure of the Ainslie wake model. Both models practically provide identical streamwise velocity fields (Figs. B1(a) and B1(b)), yet a completely different radial velocity fields (Figs.

B1(c) and B1(d)) and, in turn, a significant residual is obtained when calculating the mass conservation of the Larsen wake model (Fig. B1(e)). Therefore, we revisited the derivation of the velocity formulas from the Larsen model as follows:

$$M = \left( \frac{35}{2\pi} \right)^{3/10} (3c_1^2)^{-1/5}$$  (B1)

$$N = (3c_1^2 C_t A)^{-1/2}$$  (B2)

$$K = \frac{1}{9} (C_t A)^{1/3}$$  (B3)

$$u_x = -KN^2 r^3 x^{-\frac{5}{3}} + 2KMN r^{\frac{3}{2}} x^{-\frac{7}{6}} - KM^2 x^{-\frac{2}{3}}$$  (B4)

$$u_r = 3KN^2 r^4 x^{-\frac{8}{3}} - 6KMN r^{\frac{5}{2}} x^{-\frac{13}{6}} + 3KM^2 r x^{-\frac{5}{3}}$$  (B5)

The three contributions of the mass conservation can be written as:

$$\frac{\partial u_x}{\partial x} = \frac{5}{3} KN^2 r^3 x^{-\frac{8}{3}} - \frac{7}{3} KMN r^{\frac{3}{2}} x^{-\frac{13}{6}} + \frac{2}{3} KM^2 x^{-\frac{5}{3}}$$  (B6)

$$\frac{\partial u_r}{\partial r} = 12KN^2 r^3 x^{-\frac{8}{3}} - 15KMN r^{\frac{3}{2}} x^{-\frac{13}{6}} + 3KM^2 x^{-\frac{5}{3}}$$  (B7)

$$\frac{u_r}{r} = 3KN^2 r^3 x^{-\frac{8}{3}} - 6KMN r^{\frac{3}{2}} x^{-\frac{13}{6}} + 3KM^2 x^{-\frac{5}{3}}$$  (B8)

 

If we sum up Eqs. B6, B7, B8, the result will not be zero, which means the mass is not conserved in the model. Now we look at the derivation of $u_x$ for the Larsen wake model:

$$u_x = U_\infty (C_t A x^{-2})^{-\frac{1}{3}} \chi_1(\zeta) \tag{B9}$$

$$\chi_1(\zeta) = \chi_{11}(\xi) \tag{B10}$$

$$\chi_{11}(\xi) = -(\frac{1}{3}\xi^{\frac{3}{2}} - IC_2)^2 \tag{B11}$$

$$IC_2 = \frac{1}{3}\xi_0^{\frac{3}{2}} \tag{B12}$$

$$\xi = (3c_1^2)^{-\frac{3}{2}}\zeta \tag{B13}$$

$$\zeta = r(C_t A x)^{-\frac{1}{3}} \tag{B14}$$

$$\xi_0 = (\frac{35}{2\pi})^{\frac{1}{5}}(3c_1^2)^{-\frac{2}{15}} \tag{B15}$$

which are identical to the Eqs. (4.2.3), (4.2.10), (4.2.12), (4.2.13), (4.2.9), (4.2.2) in the original paper Larsen (1988). Substituting Eqs. B10, B11, B12, B13, B14 and B15 into Eq. B9, we confirm that the expression of $u_x$ is correctly obtained as shown

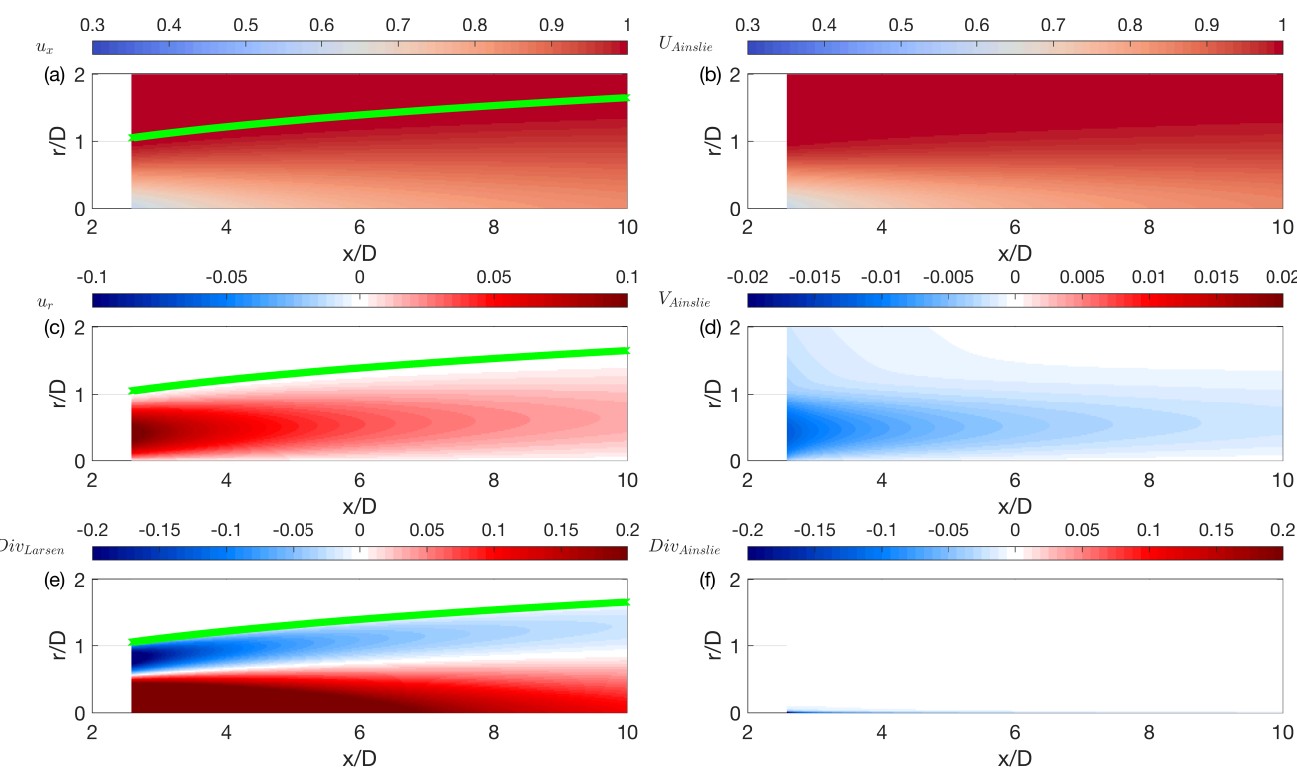

**Figure B1.** Assessment of the Larsen wake model (a) (c) and (e) against the Ainslie wake model (b), (d) and (f): (a) and (b) streamwise velocity; (c) and (d) radial velocity; (e) and (f) residual of the mass conservation. The green lines represent the wake edges defined from the Larsen wake model.





in eqn13. Regarding the radial velocity $u_r$, which is directly induced by conservation of mass as:

$$u_r = -\frac{1}{r}\int_0^r r\frac{\partial u_x}{\partial x}dr. \tag{B16}$$

We injected the Eq. B6 into Eq. B16 and the result is:

$$u_r = -\frac{1}{3}KN^2r^3x^{-\frac{8}{3}} - \frac{2}{3}KMNr^{\frac{3}{2}}x^{-\frac{16}{6}} + \frac{1}{3}KM^2x^{-\frac{5}{3}}, \tag{B17}$$

which is not equal to Eq. B5, thus we re-derived Eq. (4.2.17) in Larsen (1988) starting from:

$$u_r = \frac{U_\infty}{3}(3c_1^2C_t^2A^2)^{\frac{1}{3}}x^{-\frac{4}{3}}\xi\chi_{11}(\xi). \tag{B18}$$

We substitute the eqns B11, B12, B13, B14 and B15 into eqn B18 to finally achieve:

$$\frac{u_r}{U_\infty} = -\frac{1}{27}(C_tA(x))^{1/3}x^{-3/5}r\left\{r^{-3/2}(3c_1^2C_tA(x+x_0))^{-1/2} - (\frac{35}{2\pi})^{3/10}(3c_1^2)^{-1/5}\right\}^2 \tag{B19}$$

It is noteworthy that comparing this result with Eq. 15, the first coefficient becomes -1/27 instead of 1/3, which leads to a negative radial velocity with one order of magnitude smaller than the original calculation. The negative sign is consistent with the flow entrainment expected for the downstream recovery of a turbulent wake (Schlichting et al. , 2016). To verify divergence computed from the corrected radial velocity, we get:

$$\frac{\partial u_r}{\partial r} = -\frac{4}{3}KN^2r^3x^{-\frac{8}{3}} + \frac{5}{3}KMNr^{\frac{3}{2}}x^{-\frac{13}{6}} - \frac{1}{3}KM^2x^{-\frac{5}{3}} \tag{B20}$$

$$\frac{u_r}{r} = -\frac{1}{3}KN^2r^3x^{-\frac{8}{3}} + \frac{2}{3}KMNr^{\frac{3}{2}}x^{-\frac{13}{6}} - \frac{1}{3}KM^2x^{-\frac{5}{3}} \tag{B21}$$

It is clear that if we sum up the above equations and eqn B6, the divergence equals to zero.

## Appendix C: Numerical scheme to solve the Ainslie model

The Ainslie wake model consists of two governing equations, the continuity and momentum budgets, which are solved through the boundary layer approximation:

$$\frac{\partial u_x}{\partial x} + \frac{1}{r}\frac{\partial(ru_r)}{\partial r} = 0 \tag{C1}$$

$$u_x\frac{\partial u_x}{\partial x} + u_rV\frac{\partial u_x}{\partial r} = -\frac{1}{r}\frac{\partial(r\overline{u_xu_r})}{\partial r} \tag{C2}$$

Knowing that $-\overline{u_xu_r} = \epsilon\frac{\partial u_x}{\partial r}$, we can substitute this equation into momentum equation and discretize the result as:

$$\frac{\partial u_x^{i;j}}{\partial x} = \frac{\epsilon}{u_x^{i;j}}(\frac{1}{r}\frac{\partial u_x^{i;j}}{\partial r} + \frac{\partial^2 u_x^{i;j}}{\partial r^2}) - \frac{u_r^{i;j}}{u_x^{i;j}}\frac{\partial u_x^{i;j}}{\partial r}, \tag{C3}$$





where $i$ and $j$ are the dummy index in the $x$ and $r$ directions, respectively. $i = 1, 2, ...., Nx$ and $j = 2, 3, ..., Nr$. It is worth to mention that $Nx$ has to be sufficiently large to ensure numerical stability. To compute the axial velocity gradient, we need to know the radial velocity, $u_r$. Therefore, a parabolic approach advancing in the radial direction is applied. To this aim, the continuity equation is firstly discretized as:

$$\frac{\partial u_x^{i,j}}{\partial x} = -\frac{1}{r} \frac{r U_r^{i,j}}{\partial r} \tag{C4}$$

Then, we use it to replace the term on the left hand side of Eq. C3 producing:

$$\frac{u_r^{i,j}}{\partial r} = -\frac{\epsilon}{u_x^{i,j}} \left( \frac{1}{r} \frac{\partial u_x^{i,j}}{\partial r} + \frac{\partial^2 u_x^{i,j}}{\partial r^2} \right) + u_r^{i,j-1} \left( \frac{1}{u_x^{i,j}} \frac{\partial u_x^{i,j}}{\partial r} - \frac{1}{r} \right) \tag{C5}$$

Therefore, the radial velocity $u_r$ can advance in r direction by following:

$$u_r^{i,j} = u_r^{i,j-1} + \frac{\partial u_r^{i,j}}{\partial r} dr \tag{C6}$$

An initial value $u_r^{1,1} = 0$ is assumed as the one used in Larsen model. Finally, we insert Eq. C6 back to C3 to get $\frac{\partial u_x^{i,j}}{\partial x}$, and

apply:

$$u_x^{i+1,j} = u_x^{i,j} + \frac{\partial u_x^{i,j}}{\partial x} dx \tag{C7}$$

The solution of the velocity field is then advanced in the $x$ direction until the entire velocity field of interest is computed.

*Competing interests.* The authors declare that they have no conflict of interest.





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
