# Peer review of "Optimal tuning of engineering wake models through LiDAR measurements"

_Wind Energy Science, 2020_

## Referee Comment (RC1) · Anonymous Referee #1 · 18 Jul 2020

Review of Manuscript # wes-2020-72 Title: Optimal tuning of engineering wake models through LiDAR measurements Author(s): Lu Zhan, Stefano Letizia, and Giacomo Valerio Iungo

In this paper, the authors use the LiDAR measurements collected for individual wind turbine wakes to determine the optimal values of the tuning parameters in four different wake models. The manuscript is well written, and the problem is well defined. Determining the tuning parameters in the engineering wake model using the field data is of great interest and use for the wind-energy community. The paper can be accepted. However, there are a few issues that should be addressed before that:

1. In figure 1, the layout of the wind farm is shown. However, it is not well explained that which turbines are considered in the analysis.

2. It is not clear from the manuscript that the extent of the wake measurements behind the turbine. In particular, the range of downstream distance (x/D) that is used in the optimization of wake models should be provided.

3. In the manuscript is mentioned that "About 10,000 plan-position indicator (PPI) LiDAR scans of isolated wind turbine wakes have been processed to provide the non-dimensional average velocity fields used for this study". How are the wind-turbine wakes isolated?

4. In equation (9), the authors provided a relation for expansion rate (k) as a function of turbulent intensity (TI). Based on the results in Fig. 2, it seems that the data points are not enough to support that relationship. This point should be addressed in the paper.

5. Following the previous comment, there is no error bar in the measurement data. Error bars should be added to all figures.

6. The authors assumed that the wake growth rate solely is a function of TI. However, recent studies show the dependency of k to both TI and Ct. This point should be addressed in the paper.

7. In figure 4b, it is assumed that Ïţ is only a function of TI. However, in the original model, Ïţ is a function of Ct. The authors should elaborate on the dependency of this model coefficient to both TI and Ct. This is important since the authors report a wide range for Ct from 0.5 to 1.3 in the Gaussian model.

8. Following the previous comment, the authors showed the optimal value for Ct in the Gaussian wake model in Fig. 4c, which is larger than 1. This contradicts the 1D momentum theory. The authors should better elaborate on this point in the manuscript.

9. Fig. 5. Add the error bars to the measurement data.

10. In all figures with the discrete data (e.g., fig. 7), use dashed lines instead of full lines.
11. Line 218: the issue regarding the divergence of Eq. 12 has been solved in the following papers that can be mentioned in the manuscript.

Abkar, M., Sørensen, J.N. and Porté-Agel, F., 2018. An analytical model for the effect of vertical wind veer on wind turbine wakes. Energies, 11(7), p.1838.

Shapiro, C.R., Starke, G.M., Meneveau, C. and Gayme, D.F., 2019. A wake modeling paradigm for wind farm design and control. Energies, 12(15), p.2956.

12. It would good to provide the physics behind the optimization of the model coefficients in different wake models, and how different parameters contribute to the shape of the wake. For example, in the Gaussian wake model (Bastankhah and Porte-Agel) changing $k^*$ tries to match the width of the wakes while $C_t$ tries to match the maximum velocity deficit. Hence, the best combinations of $k^*$ and $C_t$ give the best fit to the velocity deficit profiles. You can provide a similar analysis for the other wake models.
* * *

---

## Author Comment (AC1) · 4 Aug 2020

The authors are greatly thankful to the Reviewer for insightful comments. Our replies are reported in the following. References to pages and lines are based on the revised marked-up manuscript.

**General comment**

In this paper, the authors use the LiDAR measurements collected for individual wind turbine wakes to determine the optimal values of the tuning parameters in four different wake models. The manuscript is well written, and the problem is well defined. Determining the tuning parameters in the engineering wake model using the field data is of great interest and use for the wind-energy community. The paper can be accepted.

[Figure]

However, there are a few issues that should be addressed before that

R: We thank the Reviewer for the positive feedback on our research strategy and the results achieved. We have done our best to address the comments arisen from the Reviewer.

**Specific comments**

**Point 1** In figure 1, the layout of the wind farm is shown. However, it is not well explained that which turbines are considered in the analysis.

R: We agree with the Reviewer that this detail is missing in the manuscript. At line 101, it is now added: "According to the wind farm layout and the prevalence of southerly wind directions (Fig. 1(b)), for wind directions within the sector 145° and 235°, the wakes produced by the turbines from 1 to 6 evolve roughly towards the LiDAR location, which is a favorable condition for the LiDAR to measure with close approximation the streamwise velocity through single-wake PPI scans. Furthermore, according to the layout of Fig. 1(a), for the considered wind directions, these wind turbines are not affected by upstream wakes".

**Point 2** It is not clear from the manuscript that the extent of the wake measurements behind the turbine. In particular, the range of downstream distance ($x/D$) that is used in the optimization of wake models should be provided

R: We thank the Reviewer for this comment, which is now addressed at line 121: "... the objective function of the optimization problem is the mean percentage error (PE) calculated over the measurement domain with x-coordinates between 1.25 D and 7 $D$, while r between ±1.5 D. PE is defined via...".

**Point 3** In the manuscript is mentioned that "About 10,000 plan-position indicator (PPI) LiDAR scans of isolated wind turbine wakes have been processed to provide the non-dimensional average velocity fields used for this study". How are the wind-turbine wakes isolated?

[Figure]

R: This point should be now clearer according to the revision provided for Point 1. Indeed, for the considered wind directions the turbines 1 to 6 are not affected by upstream wakes.

**Point 4** In equation (9), the authors provided a relation for expansion rate (k) as a function of turbulent intensity (TI). Based on the results in Fig. 2, it seems that the data points are not enough to support that relationship. This point should be addressed in the paper.

R: It is noteworthy that the points of Figure 2 are not single measurements but the result of the averaging process within each cluster based on incoming wind speed and turbulence intensity. Indeed, the actual number of used PPI measurements is about 10,000. To clarify this point, at line 173, it is now reported: "The parameter kopt is proportional to $TI$, even though the R-square value of 0.85 may seem quite low due to the limited number of points used for the linear fitting. However, it is noteworthy that the data reported in Fig. 2 are obtained from the mean velocity fields of each cluster of the LiDAR measurements including about 10,000 PPI scans".

**Point 5** Following the previous comment, there is no error bar in the measurement data. Error bars should be added to all figures.

R: The data reported in Fig. 2 are the results of the optimization process described in Sect. 2. Therefore, they are not statistical values that can be reported with an error bar.

**Point 6** The authors assumed that the wake growth rate solely is a function of TI. However, recent studies show the dependency of k to both TI and Ct. This point should be addressed in the paper.

R: We completely agree with this statement. At line 179, it is reported "This different variability of $k_{opt}$ with TI indicates that this model parameter reflects not only effects of the incoming turbulence intensity on the wake evolution, but of the wakegenerated turbulence as well". At line 216, it is reported for the Bastankhah model "a secondary trend with the rotor thrust coefficient is observed, which is an effect of the wake-generated turbulence. Indeed, the operative conditions with $U_{hub}^* > 0.9$ are characterized by a slightly smaller $k_{opt}$ ".

**Point 7** In figure 4b, it is assumed that $\epsilon_{opt}$ is only a function of TI. However, in the original model, $\epsilon_{opt}$ is a function of Ct. The authors should elaborate on the dependency of this model coefficient to both TI and Ct. This is important since the authors report a wide range for Ct from 0.5 to 1.3 in the Gaussian model.

R: We thank the Reviewer for raising this important comment that deserves further discussions. Indeed, the parameter Ïţ of the Bastankhah model is, by definition, only a function of Ct ($\epsilon = 0.2\sqrt{\beta}$, where $\beta = 1/2(1 + \sqrt{(1 + C_t)}/\sqrt{(1 - C_t)}$, as it is now reported at line 207. However, already in Bastankhah et al 2014, it was noticed that varying Ct between 0.42 to 0.8, $\epsilon$ only changes from 0.219 up to 0.257. Our study on optimization of the model parameters based on wake LiDAR measurements confirms that Ïţ has small variations with Ct, while the main variability is connected with TI, which might be an effect of the modulation on wake recovery induced by the atmospheric stability. In the manuscript at line 225, it is now reported: "The offset of the standard deviation of the velocity profile, Ïţ, which is by definition only a function of Ct, slightly decreases with increasing $U_{hub}^*$, suggesting that lower Ct values associated with high $U_{hub}^*$, i.e. for operations in region 3 of the power curve, lead to a narrower and shallower velocity deficit in the near wake. However, the results of the model optimization show that the main variability of Ïţ is connected with the incoming turbulence intensity and, specifically, $\epsilon$ decreases with increasing TI. This effect on Ïţ might be due to the modulation induced on wake recovery by the atmospheric stability".

**Point 8** Following the previous comment, the authors showed the optimal value for Ct in the Gaussian wake model in Fig. 4c, which is larger than 1. This contradicts the 1D momentum theory. The authors should better elaborate on this point in the manuscript.
R: We thank the Reviewer for this important comment. We have emphasized that the

Ct obtained through the 1D momentum theory significantly underestimates the respective value obtained by applying the streamwise momentum budget to the LiDAR data or the respective value obtained through the wake-model optimization. These results confirmed results from a previous work (Iungo et al. 2018b), where we leveraged LiDAR measurements and a RANS solver to prove that the thrust coefficient of a wind turbine, so that associated with a wind turbine wake, is generally larger to that predictable through the 1D momentum theory and, in turn, to that associated with the power generation. Indeed, the wake of a wind turbine is the result of the thrust force due to the transformation of the wind kinetic energy in the mechanical rotation energy of the wind turbine, but also includes the drag connected with the bluff body behavior of the nacelle, the tower and blade stall. At line 235, it is reported: "In Fig. 3, the optimized thrust coefficient, Ctopt, is generally higher than the respective values predictable with the 1D stream-tube assumption (Eq. 5), because including contributions to drag due to the bluff-body behavior of the turbine tower, nacelle and blade stall".

**Point 9** Fig. 5. Add the error bars to the measurement data.

R: Based on the ensemble averaging and clustering analysis of the LiDAR data (Zhan et al., 2019), the standard error on the weighted mean is always lower than 0.8%, as now reported at line 111. If we would add the error bars, the plot will result to be more confusion without adding significant information, as shown in the figure attached.

**Point 10** In all figures with the discrete data (e.g., fig. 7), use dashed lines instead of full lines.

R: Figs. 2, 4, 7, 8 have been revised accordingly.

**Point 11** Line 218: the issue regarding the divergence of Eq. 12 has been solved in the following papers that can be mentioned in the manuscript. Abkar, M., Sørensen, J.N. and Porté-Agel, F., 2018. An analytical model for the effect of vertical wind veer on wind turbine wakes. Energies, 11(7), p.1838. Shapiro, C.R., Starke, G.M., Meneveau, C. and Gayme, D.F., 2019. A wake modeling paradigm for wind farm design and control.

Energies, 12(15), p.2956.

R: These papers are now cited in the manuscript.

**Point 12** It would good to provide the physics behind the optimization of the model coefficients in different wake models, and how different parameters contribute to the shape of the wake. For example, in the Gaussian wake model (Bastankhah and Porte-Agel) changing $k^*$ tries to match the width of the wakes while Ct tries to match the maximum velocity deficit. Hence, the best combinations of $k^*$ and Ct give the best fit to the velocity deficit profiles. You can provide a similar analysis for the other wake models.

R: We have added comments on the physical contribution of the model parameters to the characteristics of the wakes throughout the manuscript. For instance, for the Jensen model, at line 159 it is reported "The wake expansion coefficient, k, is defined in analogy with the jet spreading within shear flows (Pope, 2000)". For the Bastankhah model at line 222 "(Eq.11), $\epsilon$, which is by definition only a function of Ct, slightly decreases with increasing $U_{hub}^*$, suggesting that lower Ct values associated with high $U_{hub}^*$, i.e. for operations in region 3 of the power curve, lead to a narrower and shallower velocity deficit in the near wake". For the Larsen model at line 287 "As mentioned in section 3.3, $x_0$ is defined as the distance between the rotor position and the origin of the used coordinate system. Nonetheless, it can also be denoted physically as the position where the initial wake width equals to one rotor diameter. Therefore, a faster wake recovery rate due to higher incoming turbulence makes this condition occurring closer to the turbine rotor" or at line 291 "Regarding the wake recovery rate reported in Fig. 7(c), we can see the enhancement of turbulent mixing as a function of increasing turbulence intensity, which can be modeled through a linear function with a slope of 0.68 and interception of 0.01". At line 321 for the Ainslie model "the wake-generated turbulence is taken into account through the parameters $k_l$ and $K_M$".

[Figure]

Please also note the supplement to this comment:
https://wes.copernicus.org/preprints/wes-2020-72/wes-2020-72-AC1-supplement.pdf

[Figure]

[Figure]

**Fig. 1.** Figure Point 9

**Supplement:**

[revised manuscript text omitted]

---

## Referee Comment (RC2) · Anonymous Referee #2 · 6 Aug 2020

Review of "Optimal tuning of engineering wake models through LiDAR measurements"

**Summary:** Input parameters of four engineering wake models are optimized by finding the set of model parameters that provide the smallest errors between the modeled wake velocity field and cluster-averaged LiDAR measurements of the wake velocity field. The conclusions concern the strengths and weaknesses of the four compared models.

**General comments:** The research goals are relevant to field and motivated in the introduction. The methods are missing some essential information. Apart from the open questions arising from this missing information, the optimization and the results are presented in a clear and comprehensible way. The conclusions could be improved with recommendations for the application of the models. My main comments are (details in the specific comments below):

1) The manuscript is missing information on the Doppler LiDAR measurements and their processing.
2) The Jenzen model and the Basthankhah model predict the normalized velocity deficit of the wake (Eq. 6 and Eq. 12). The optimization uses the longitudinal velocity (Eq. 1). Therefore, I believe an inflow velocity profile has to enter the optimization at one point, but the manuscript does not provide information on this.
3) It is not clear which spatial volume is used to compute the error between the model and the LiDAR measurements and, therefore, it is difficult to assess if neighboring wind turbine wakes or offsets of the wake center position might affect the error unintentionally.
4) It might be interesting if the conclusions could elaborate on the following questions: What are the benefits of optimally calibrated models compared to using some of the general assumptions for the parameters found in literature? How transferable are the results of this optimization to other sites?

Overall, I recommend considering the manuscript for publication after the authors have addressed those points.

**Specific comments:**

- Figure 1: Panel (a) could use a scale and what is the time period used for the plot in panel (b)? Does it correspond to the data used in the results?
- Lines 93-99: Information is missing for the Doppler LiDAR: What was the elevation angle of the PPI scans? What was the azimuth step? How much time does each scan take? And how are the wake centered for the comparison with the model? A brief summary of the data processing would be helpful, even if it is described in detail by Zhan et al. (2019).
- Eq. (1): Are the wake measurements of the LiDAR processed such that the wake is centered in the spanwise plane? Otherwise the error will include contributions from a different wake positions. And which downwind distances are used to compute the error?
- Eq. (1), Eq. (6), and Eq. (12): From Eq. (1) it seems that the model prediction of the mean longitudinal velocity field is compared with cluster-average from the Doppler LiDAR for the optimization. However, the Jensen model and the Bastankhah model predict the normalized velocity deficit in Eq. (12). To compute the longitudinal velocity field from the model, an inflow velocity profile is required. Therefore, the following things are unclear to me: where does inflow profile come from? Is it used to normalize the LiDAR measurements or combined with the model? Does it contribute to the model error?
- Figures 2a, 7a, and 7b: Some of the optimization clusters seem to hit a threshold (e.g. the optimization of the wake growth rate seems to plateau at 0.1 in Fig. 2a). Is there an explanation for this or could it be a too small search space of the optimization by mistake?

- Line 308 and Figure 8: I have difficulties to relate the mentioned peak at 7% with the shown data. It seems that only two of the clusters have a peak and most not.
- Figure 9: The caption should state that a normalized velocity is shown and that the green lines are the wake edge. Would it make sense to use $\sigma$ as the wake width in case of the Basthankhah model?
- Figure 11: Which spatial volume is used for the computation of the percentage error? If large areas outside of the wake are used, then the error might also reflect undesired effects (e.g. neighbouring wind turbine wakes or inhomogeneous wind fields outside of the wake). Since the models only make predictions of the wake, it would be most sensible to use only the wake for the computation of the error.
- Conclusions: I am wondering what is the gain in error reduction with optimally calibrated models compared to using frequently made assumptions in literature? For example how much lower is the error of the calibrated Jensen model or Basthankhah model compared to using wake growth rate assumptions provided by Fuertes et al. (2018) or Peña et al. (2015)? Alternatively, it might be interesting to investigate the error as a function of the model parameters to gain insights into the sensitivity of the error to the parameters.

**Technical comments:**

- Line 67: I believe the abbreviation SCADA was not introduced yet.
- Line 88: The "while" in this sentence seems odd, because the topography data and meteorological data have no connection with each other.
- Line 89: Remove space before the comma.
- Line 95: Should be "of" instead of "on".
- Line 230: Remove the "e" before "Bastankhah wake model".
- Line 245: The citation should be "Larsen et al. (2003)" instead of "(Larsen et al., 2003)".

References

Peña, Alfredo, Pierre-Elouan Réthoré, and M. Paul van der Laan. "On the application of the Jensen wake model using a turbulence-dependent wake decay coefficient: the Sexbierum case." Wind Energy 19.4 (2016): 763-776.

Carbajo Fuertes, Fernando, Corey D. Markfort, and Fernando Porté-Agel. "Wind turbine wake characterization with nacelle-mounted wind lidars for analytical wake model validation." Remote sensing 10.5 (2018): 668.

---

## Author Comment (AC2) · 25 Aug 2020

**Reply to the comments provided by the Anonymous Referee #2 on the manuscript wes-2020-72 entitled "Optimal tuning of engineering wake models through LiDAR measurements", by L. Zhan, S. Letizia and G. V. Iungo**

The authors are greatly thankful to the Reviewer for insightful comments. Our replies are reported in the following. References to pages and lines are based on the revised marked-up manuscript.

**General comments**

*The research goals are relevant to field and motivated in the introduction. The methods are missing some essential information. Apart from the open questions arising from this missing information, the optimization and the results are presented in a clear and comprehensible way. The conclusions could be improved with recommendations for the application of the models. My main comments are (details in the specific comments below):... Overall, I recommend considering the manuscript for publication after the authors have addressed those points.*

R: We thank the Reviewer for the positive feedback on our research strategy and the results achieved. We have added more comments and suggestions on the applications of the models following the Reviewer's comments.

**1.** *The manuscript is missing information on the Doppler LiDAR measurements and their processing.*

R: We agree that more details on the LiDAR measurements are needed for the sake of clarity, rather than only referring to Zhan *et al.* 2019. As reported at the Specific Comment 2, more information is now added to the text.

**2.** *The Jensen model and the Basthankhah model predict the normalized velocity deficit of the wake (Eq. 6 and Eq. 12). The optimization uses the longitudinal velocity (Eq. 1). Therefore, I believe an inflow velocity profile has to enter the optimization at one point, but the manuscript does not provide information on this.*

R: The Reviewer is right. This step of the post-processing of the LiDAR data was not described in detail and only referred to the previous paper Zhan *et al.* 2019. The calculation of the non-dimensional velocity field is now reported in the text (see Specific Comment 2).

**3.** *It is not clear which spatial volume is used to compute the error between the model and the LiDAR measurements and, therefore, it is difficult to assess if neighboring wind turbine wakes or offsets of the wake center position might affect the error unintentionally.*

R: The LiDAR scanning strategy was designed to only probe isolated wind turbines and to avoid wake interactions. At line 101, it is now reported: "According to the wind farm layout and the prevalence of southerly wind directions (Fig. 1), for wind directions within the sector 145° and 235°, the wakes produced by the turbines from 1 to 6 evolve roughly towards the LiDAR location, which is a favorable condition for the LiDAR to measure with close approximation the streamwise velocity through single-wake plan-position indicator (PPI) scans. Furthermore, according to the layout of Fig. 1(a), for the considered wind directions, these wind turbines are not affected by upstream wakes". Regarding the wake region considered for the optimal tuning of the model parameters, it is now reported at line 131: "… the mean percentage error (*PE*) calculated over the measurement domain with *x*-coordinates between $1.25\,D$ and $7\,D$, while *r* between $\pm 1.5\,D$".

**4.**     *It might be interesting if the conclusions could elaborate on the following questions: What are the benefits of optimally calibrated models compared to using some of the general assumptions for the parameters found in literature? How transferable are the results of this optimization to other sites?*

R: For this project, we considered a typical utility-scale wind farm on flat terrain, with a typical daily cycle of the atmospheric stability for onshore sites. Therefore, we believe that the value obtained for the model parameters can be generalized to other onshore wind projects not affected by significant topography wind distortion. In the Conclusions at L 455, it is now reported: "The optimal tuning of the considered wake models has enabled to significantly reduce the mean percentage error in the predictions of the wake velocity field. For certain clusters of the LiDAR dataset, the mean percentage error has been four times smaller than for the respective baseline wake prediction obtained by using standard parameter values available from the literature. Considering that the wind farm under investigations is characterized by a typical layout, flat terrain and typical daily cycle of the atmospheric stability for onshore wind farms, we expect that similar improvements in wake-prediction accuracy can be generally achieved for wind farms with similar characteristics by using the reported optimally-tuned model parameters". Fig. 11 has been significantly revised by adding a direct comparison of the percentage error (PE) between the wake predictions obtained with the standard model parameters and the predictions obtained with the optimally-tuned models. The text at L 383-402 describes in detail the significant improvements achieved through the optimal calibration of the wake models.

**Specific comments**

**1.**     *Figure 1: Panel (a) could use a scale and what is the time period used for the plot in panel (b)? Does it correspond to the data used in the results?*

R: A scale is now added to Fig. 1(a). In the caption of Fig. 1, it is now reported that the wind data used to calculate the wind rose were collected for the entire duration of the LiDAR experiment.

**2.**     *Lines 93-99: Information is missing for the Doppler LiDAR: What was the elevation angle of the PPI scans? What was the azimuth step? How much time does each scan take? And how are the wake centered for the comparison with the model? A brief summary of the data processing would be helpful, even if it is described in detail by Zhan et al. (2019).*

R: At line 106, it is now reported: "The LiDAR measurements were typically performed by using a range gate of 50 m, elevation angle of $\phi=3°$, azimuthal range of 20°, rotation speed of the scanning head of 2°/s, leading to a typical scanning time for a single PPI of 10 s. After rejecting LiDAR data with a carrier-to-noise ratio (CNR) lower than -25 dB, a proxy for the streamwise velocity is obtained through the streamwise equivalent velocity, $U_{eq} = V_r/[\cos\phi\cos(\theta - \theta_w)]$, where $\theta$ is the azimuthal angle of the LiDAR laser beam and $\theta_w$ is the wind direction. The streamwise equivalent velocity is then made non-dimensional through the velocity profile in the vertical direction of the incoming boundary layer. The latter is estimated for each PPI scan through the 70-th percentile of $U_{eq}$ at each height. In Zhan *et al.* 2019, it was shown how this technique allows to remove turbulent gusts and LiDAR samples with reduced wind speed in correspondence of the wind-turbine wake. The reference frame used has *x*-direction aligned with the wake direction, which is estimated with linear fitting of the wake centers at various downstream locations. The transverse position of the wake center is defined as the location of the minimum

velocity obtained by fitting the velocity data at a specific downstream location through a Gaussian function".

**3.**     *Eq. (1): Are the wake measurements of the LiDAR processed such that the wake is centered in the spanwise plane? Otherwise the error will include contributions from a different wake positions. And which downwind distances are used to compute the error?*
R: These details of the post-processing of the LiDAR data are now added to the text. At line 111, it is reported: "The reference frame used has $x$-direction aligned with the wake direction, which is estimated with linear fitting of the wake centers at various downstream locations. The transverse position of the wake center is defined as the location of the minimum velocity obtained by fitting the velocity data at a specific downstream location through a Gaussian function". At line 131: "… the mean percentage error (*PE*) calculated over the measurement domain with $x$-coordinates between 1.25 $D$ and 7 $D$, while $r$ between $\pm 1.5$ $D$".

**4.**     *Eq. (1), Eq. (6), and Eq. (12): From Eq. (1) it seems that the model prediction of the mean longitudinal velocity field is compared with cluster-average from the Doppler LiDAR for the optimization. However, the Jensen model and the Bastankhah model predict the normalized velocity deficit in Eq. (12). To compute the longitudinal velocity field from the model, an inflow velocity profile is required. Therefore, the following things are unclear to me: where does inflow profile come from? Is it used to normalize the LiDAR measurements or combined with the model? Does it contribute to the model error?*
R: At line 110, it is now reported: "The streamwise equivalent velocity is then made non-dimensional through the velocity profile in the vertical direction of the incoming boundary layer. The latter is estimated for each PPI scan through the 70-th percentile of $U_{eq}$ at each height. In Zhan *et al.* 2019, it was shown how this technique allows to remove turbulent gusts and LiDAR samples with reduced wind speed in correspondence of the wind-turbine wake".

**5.**     *Figures 2a, 7a, and 7b: Some of the optimization clusters seem to hit a threshold (e.g. the optimization of the wake growth rate seems to plateau at 0.1 in Fig. 2a). Is there an explanation for this or could it be a too small search space of the optimization by mistake?*
R: We thank the Reviewer for this important comment. For Fig. 2(a), the wake expansion coefficient of the Jensen model, $k$, is varied between 0.001 and 0.3 (L 181). Therefore, the maximum value of about 0.1 is significantly below its upper limit. For Fig. 7(b), the maximum value of $x_0$ is 3.01, and indeed several cases hit the upper limit. However, this limit has been chosen based on the physical interpretation of $x_0$, which is the streamwise offset for the wake evolution, which should occur in the near wake. For the thrust coefficient of the Larsen model, which is reported in Fig. 7(a), an upper limit of 1 was set in the first version of the manuscript based on the model constraint connected with Eq. A3. However, considering that for the optimal tuning of the Larsen model, $Ct$ is a free parameter, then the maximum value has been increased to 1.5. At L 287, it is now reported: "The thrust coefficient, $C_t$, is varied between 0.4 and 1.5 with a step of 0.01, $c_1$ optimal value is searched from 0.01 up to 0.25 with a resolution of 0.002, while $x_0$ ranges from 0.01 up to 3.01 with a step of 0.05. It is noteworthy that $Ct$ values larger than 1 are allowed since the constraint of Eq. A3 is bypassed by considering $Ct$ as a free input parameter". As shown in Fig. 7(a), several data clusters owing to region 2 of the power curve now achieve $Ct$ larger than 1. Figs. 3, 6(a), 7, 10, and 11 have been updated accordingly.

**6.**     *Line 308 and Figure 8: I have difficulties to relate the mentioned peak at 7% with the shown data. It seems that only two of the clusters have a peak and most not.*
R: This is now rephrased as (L 344): "A region with higher $k_l$ is observed for $TI<15\%$, then $kl$ approaches zero for higher $TI$ values".

**7.**     *Figure 9: The caption should state that a normalized velocity is shown and that the green lines are the wake edge. Would it make sense to use $\sigma$ as the wake width in case of the Basthankhah model?*
R: In Fig. 9(c), the wake edge is reported in correspondence of $2\sigma$, which includes 95% of the total momentum deficit (Aitken et al. JTECH 2014). Caption of Fig. 9 is now: "Normalized velocity for the cluster with $U_*$ of [0.76, 0.81] and $TI$ of [13.5%, 19.4%] : (a) LiDAR data; (b) Jensen wake model; (c) Bastankhah wake model; (d) Larsen wake model; (d) Ainslie wake model. Green lines represent the wake edges, while for (c) they represent the spanwise position corresponding to $2\sigma$".

**8.**     *Figure 11: Which spatial volume is used for the computation of the percentage error? If large areas outside of the wake are used, then the error might also reflect undesired effects (e.g. neighbouring wind turbine wakes or inhomogeneous wind fields outside of the wake). Since the models only make predictions of the wake, it would be most sensible to use only the wake for the computation of the error.*
R: The Reviewer is right, the wake region where the *PE* is calculated is now specified in the manuscript. At L 131, it is now reported: "… the mean percentage error (*PE*) calculated over the measurement domain with *x*-coordinates between 1.25 *D* and 7 *D*, while *r* between ±1.5 *D*".

**9.**     *Conclusions: I am wondering what is the gain in error reduction with optimally calibrated models compared to using frequently made assumptions in literature? For example how much lower is the error of the calibrated Jensen model or Basthankhah model compared to using wake growth rate assumptions provided by Fuertes et al. (2018) or Peña et al. (2015)? Alternatively, it might be interesting to investigate the error as a function of the model parameters to gain insights into the sensitivity of the error to the parameters.*
R: We thank the Reviewer for this important comment. We have now performed a direct comparison between the baseline wake predictions, which are obtained with the typical model parameters available from literature, and the respective ones obtained through the optimally-tuned parameters in terms of percentage error. This analysis is reported in the revised Fig. 11 and the text at L 382-404. We quantify the improvements in the model accuracy, which for certain clusters of the LiDAR dataset entails a reduction of the percentage error of about four times.

**Technical comments**

**1.**     *Line 67: I believe the abbreviation SCADA was not introduced yet.*
R: It is now added.

**2.**     *Line 88: The "while"in this sentence seems odd, because the topography data and meteorological data have no connection with each other.*
R: Revised.

**3.** *Line 89: Remove space before the comma.*
R: Revised.

**4.** *Line 95: Should be "of"instead of "on".*
R: Revised.

**5.** *Line 230: Remove the "e" before "Bastankhah wake model".*
R: Revised.

**6.** *Line 245: The citation should be "Larsen at al. (2003)" instead of (Larsen at al., 2003)".*
R: Revised.